# Identification of long non-coding RNAs in advanced prostate cancer associated with androgen receptor splicing factors

Ken-ichi Takayama[1], Tetsuya Fujimura[2], Yutaka Suzuki[3] & Satoshi Inoue [1,4✉]

The molecular and cellular mechanisms of development of castration-resistant prostate cancer (CRPC) remain elusive. Here, we analyzed the comprehensive and unbiased expression profiles of both protein-coding and long non-coding RNAs (lncRNAs) using RNA-sequencing to reveal the clinically relevant molecular signatures in CRPC tissues. For protein-coding genes upregulated in CRPC, we found that mitochondria-associated pathway, androgen receptor (AR), and spliceosome associated genes were enriched. Moreover, we discovered AR-regulated lncRNAs, *CRPC-Lncs*, that are highly expressed in CRPC tissues. Notably, silencing of two lncRNAs (*CRPC-Lnc #6: PRKAG2-AS1* and *#9: HOXC-AS1*) alleviated CRPC tumor growth, showing repression of AR and AR variant expression. Mechanistically, subcellular localization of the splicing factor, U2AF2, with an essential role in AR splicing machinery was modulated dependent on the expression level of *CRPC-Lnc #6*. Thus, our investigation highlights a cluster of lncRNAs which could serve as AR regulators as well as potential biomarkers in CRPC.

[1] Department of Systems Aging Science and Medicine, Tokyo Metropolitan Institute of Gerontology, 35-2 Sakae-cho, Itabashi-ku, Tokyo 173-0015, Japan. [2] Department of Urology, Jichi Medical University, Shimotsuke, Tochigi 329-0498, Japan. [3] Department of Computational Biology and Medical Sciences, Graduate School of Frontier Sciences, The University of Tokyo, Chiba 277-8562, Japan. [4] Division of Gene Regulation and Signal Transduction, Research Center for Genomic Medicine, Saitama Medical University, Hidaka, Saitama 350-1241, Japan. ✉email: sinoue@tmig.or.jp

Prostate cancer is the most common cancer in men and is the second leading cause of cancer death in the developed countries[1]. Advances in the treatment of prostate cancer by surgery and radiation have contributed to the favorable prognosis for patients[2]. In addition, androgen-deprivation therapy (ADT) has resulted in significantly prolonged survival of patients with metastatic prostate cancer. Drugs used in ADT including enzalutamide, bicalutamide, and abiraterone inhibit androgen signaling, which is the major growth-promoting pathway mediated by the androgen receptor (AR). Despite the initial success of the treatment, many aggressive cancers, termed castration-resistant prostate cancer (CRPC), grow even in castrate levels of androgen. The main cause of CRPC development is the reactivation of AR signaling[3]. Previous studies have discovered the underlying mechanisms for the enhanced AR downstream signaling, expression of AR, and its splice variants lacking the ligand-binding domain (ARVs) in CRPC[4]. The variant AR-V7 was shown to regulate distinct and androgen-independent activation of its downstream signals, which contributes to the development of CRPC[5,6]. Thus investigation of the molecular mechanisms of activation of AR, AR-V7, and its downstream signals would have a significant clinical impact in treating CRPC.

Long non-coding (lnc) RNAs have diverse functions such as epigenetic and gene regulation in cancer cells[7], including prostate cancer cells[8]. Previous reports highlighted lncRNA-mediated association of RNA-binding proteins or transcription factors with specific genomic regions in prostate cancer progression[9]. We previously reported that an androgen-induced lncRNA, CTBP1-AS, in the antisense region of carboxyl terminal binding protein 1 (CTBP1) promotes castration-resistant tumor growth[10]. CTBP1-AS modulates the global epigenetic status to repress negative cell cycle regulators or AR corepressor, CTBP1, to promote tumor growth and activation of AR activity. Interestingly, recent reports showed that androgen-regulated lncRNAs are implicated in several processes in AR activation. Androgen-repressed HOTAIR increased AR expression through posttranslational stabilization of protein by blocking the E3 ligase, MDM2, targeting AR for ubiquitylation[11]. Androgen-induced ARLNC1 is highly expressed in CRPC tissues and stabilizes the AR mRNA by RNA–RNA hybridization to enhance AR expression level posttranscriptionally[12]. Thus lncRNAs, particularly the AR-regulated lncRNAs, form an important regulatory layer in global gene expression. Moreover, alterations of the lncRNA expression profile in CRPC are assumed to be one of driving forces for cellular transformation. However, the clinical relevance of lncRNA expression and associated molecular mechanisms in CRPC has not been fully understood.

Transcriptional characterization of cancer tissues can reveal important molecular signatures associated with the disease progression[8,12]. In the previous study, we measured the expression levels of targeted protein-coding genes using tumor samples from patients with metastasis and demonstrated that hormone-regulated and stem cell-related markers could predict survival of these patients[13]. Meanwhile, more comprehensive and unbiased analyses of gene expression are preferable in tumors to identify the clinical and molecular signatures responsible for the aggressiveness of prostate cancer. Here we performed directional RNA-sequencing (RNA-seq) using clinical samples obtained from localized prostate cancer and CRPC patients. For the protein-coding genes, we found a cluster of upregulated genes in CRPC. In addition, by integrating with AR chromatin immunoprecipitation-sequencing (ChIP-seq) data that we performed using several prostate cancer cell lines[14–20], we found changes in the AR program in CRPC tissues. Furthermore, we discovered a cluster of CRPC-enriched lncRNAs (abbreviated as CRPC-Lncs), which are regulated by AR. Using functional analysis, we revealed that splicing factors with an

essential role in AR splicing machinery associate with this cluster of lncRNAs, indicating that these RNAs can create the conditions for promoting AR splicing and overexpression. Taken together, our investigation identified a cluster of lncRNAs, CRPC-Lncs, which could serve as new biomarkers and therapeutic targets in CRPC.

## Results

**Gene expression signatures associated with CRPC development.** To determine the functionally relevant gene expression in prostate cancer progression, we performed RNA-seq analysis of six benign prostate, eight localized prostate cancer, and six CRPC tissue samples. Using the Bowtie/TopHat-processed[15,18] RNA-seq data, we mapped the genome-wide expression profile of protein-coding genes and lncRNAs registered in RefSeq, GENCODE, and NONCOE databases. We then compared the gene expression levels of CRPC tissues with benign prostate and localized prostate cancer tissues, demonstrating that CRPC-associated alteration of signals in tumors occurred on a genome-wide scale (Fig. 1a).

To explore the signals involved in prostate cancer progression, we identified transcripts upregulated in CRPC samples compared to both benign prostate and localized prostate cancer tissues and named this category as "Type_A" genes (total of 2187 genes), which are assumed to be abundant in CRPC specifically (Fig. 1b, c). We then made a heatmap of the highly expressed Type_A genes that include representative CRPC marker genes such as PCAT1[21], TRIM25/Efp[22], EZH2[23,24], UBE2C[25], and AR[3] (Fig. 1b). Thus this category includes CRPC-specific gene signatures. In addition, we also identified transcripts significantly upregulated in CRPC compared to only benign prostate tissues and then excluded genes found in Type_A. We named this category as "Type_B" and found that genes classified in this category were generally highly expressed in both CRPC and localized prostate cancer, suggesting that these genes might be important in both phases of the disease progression (Supplementary Fig. 1a, b). Interestingly, Type_B genes included previously reported AR-regulated genes such as AMACR[26], ARLNC1, CTBP1-AS, and COBLL1[15] that are important in CRPC development (Supplementary Fig. 1a, b). These results suggest that genes to be important in CRPC progression could be divided into two categories. Then we compared our RNA-seq results with the microarray data from two previous studies[27,28] available through Gene Expression Omnibus. We analyzed how the genes upregulated in CRPC tissues (Type_A) were expressed in each of these datasets. We found that most of them are similarly upregulated in the metastatic CRPC tissues based on these microarray data, in line with our RNA-seq analysis (Fig. 1d).

Next, we analyzed the molecular functions upregulated in CRPC tissues by analyzing Kyoto Encyclopedia of Genes and Genomes pathway. Among RefSeq genes in Type_A (total of 1679 genes), we found remarkable enrichment of genes involved in biological activities, such as oxidative phosphorylation (OXPHOS), spliceosome, DNA replication, and cell cycle (Fig. 1e). Interestingly, among the Type_B genes (total of 1853 genes) we also observed an enrichment of genes involved in RNA splicing, OXPHOS, and cell cycle (Supplementary Fig. 1c, d).

**CRPC-specific AR transcriptional program.** Because enhanced AR expression and hypersensitivity are important in the progression of CRPC, we next investigated how AR-regulated signals were enriched among CRPC-specific gene signatures. We previously performed AR ChIP-seq to map the genome-wide occupancy of AR in prostate cancer cells[14–16]. We used three AR ChIP-seq data to select AR-binding genes, which were the closest genes to AR-binding sites in androgen-dependent-type prostate cancer cells (LNCaP and VCaP) and CRPC model cells (22Rv1) (Fig. 2a). In addition, to determine the regulation of gene expression by androgen, RNA-seq studies for these cells were

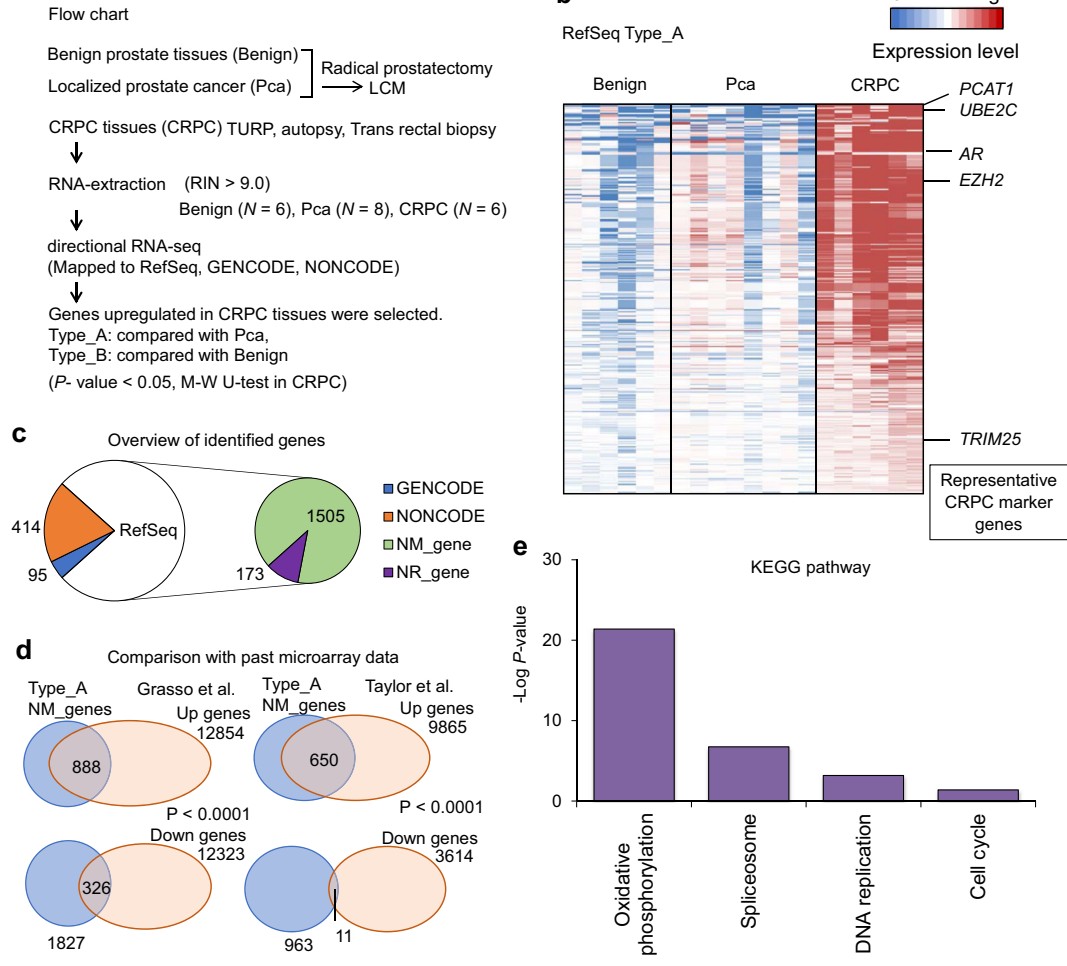

**Fig. 1 Overview of transcriptome composition and study design. a** Schematic depicting workflow of our RNA-seq study. RNA samples were obtained from prostate cancer patients by radical prostatectomy, TURP, autopsy, and transrectal biopsy. Benign prostate (Benign, $N = 6$), localized prostate cancer (Pca, $N = 8$), and castration-resistant prostate cancer (CRPC, $N = 6$) tissues including two metastatic samples (lymph node and liver) were used for directional RNA-sequencing (RNA-seq) study. We selected highly expressed genes of RefSeq (RPKM > 5), GENCODE (RPKM > 1), and NONCODE (RPKM > 1) in CRPC tissues. Genes upregulated in CRPC tissues were classified into two groups (Type_A: upregulated in CRPC tissues compared with Pca, Type_B: upregulated in CRPC tissues compared with benign. Mann–Whitney (M-W) $U$ test was performed to compare gene expression levels. $P < 0.05$ was considered to be significant. LCM laser capture microdissection, TURP transurethral resection of the prostate, RIN RNA integrity number, RPKM reads per kilobase of exon per million mapped reads. **b** Identification of genes involved in CRPC development. Expression level relative to benign prostate tissue is visualized as heatmap. RefSeq genes of Type_A, which are upregulated between CRPC and localized Pca significantly ($P < 0.05$). The genes that were previously reported to be associated with prostate cancer progression are indicated as representative CRPC marker genes. **c** Classification of the identified genes in three gene databases. We used RefSeq, GENCODE, and NONCODE databases to map sequenced tags for annotated regions. The numbers of identified Type_A genes were shown. NM_genes RefSeq genes with NM_accession numbers (protein-coding genes), NR_genes RefSeq genes with NR_accession numbers (non-coding RNA). **d** Comparison of upregulated genes in publicly available microarray datasets. Two microarray datasets registered in GEO were used[27,28] to examine whether Type_A protein-coding genes were upregulated (Up genes) or downregulated (Down genes) in metastatic CRPC tissues compared with localized prostate tissues in other cohorts. Up genes were significantly enriched with Type_A genes ($P < 0.0001$, chi-square test). **e** Kyoto Encyclopedia of Genes and Genomes (KEGG) pathway analysis of Type_A protein-coding genes.

performed. We analyzed androgen-dependent gene induction in LNCaP and VCaP by treating the cells with 10 nM dihydrotestosterone (DHT)[18]. In contrast, 22Rv1 cells were treated with siAR or control small interfering RNA (siRNA) to investigate AR-dependent signals because 22Rv1 cells express ligand-independent AR variant (AR-V7) at a high level and increased binding of AR was observed without DHT treatment in ChIP-seq data[15,16]. Then we integrated the AR ChIP-seq data with RNA-seq to identify AR-regulated genes. A total of 1748 genes were found to be induced by AR in LNCaP cells, and around half of them were also AR-induced genes in VCaP cells (Fig. 2a). In 22Rv1 cells, we found a total of 717 genes that were positively regulated by AR binding. Next, we extended our findings to

comprehensively evaluate the mechanism through which the AR-regulated genes in prostate cancer model cells were involved in the clinical course of prostate cancer (Fig. 2b, Supplementary Fig. 2a–c). Among the AR-regulated genes in both LNCaP and VCaP (LNCaP/VCaP), the number of genes significantly upregulated ($P < 0.05$) in prostate cancer tissues compared to benign prostate tissues was high (13.8%). However, most of them (10.2%) were downregulated in the CRPC tissues compared to the prostate cancer tissues (Fig. 2b, c). In contrast, among AR-regulated genes in 22Rv1 cells, the number of genes upregulated in prostate cancer tissues (8.4%) was lower ($P = 0.001$ by chi-square test) than that in LNCaP/VCaP (13.8%) (Fig. 2b), while the number of genes upregulated in CRPC tissues was relatively high (4.5%)

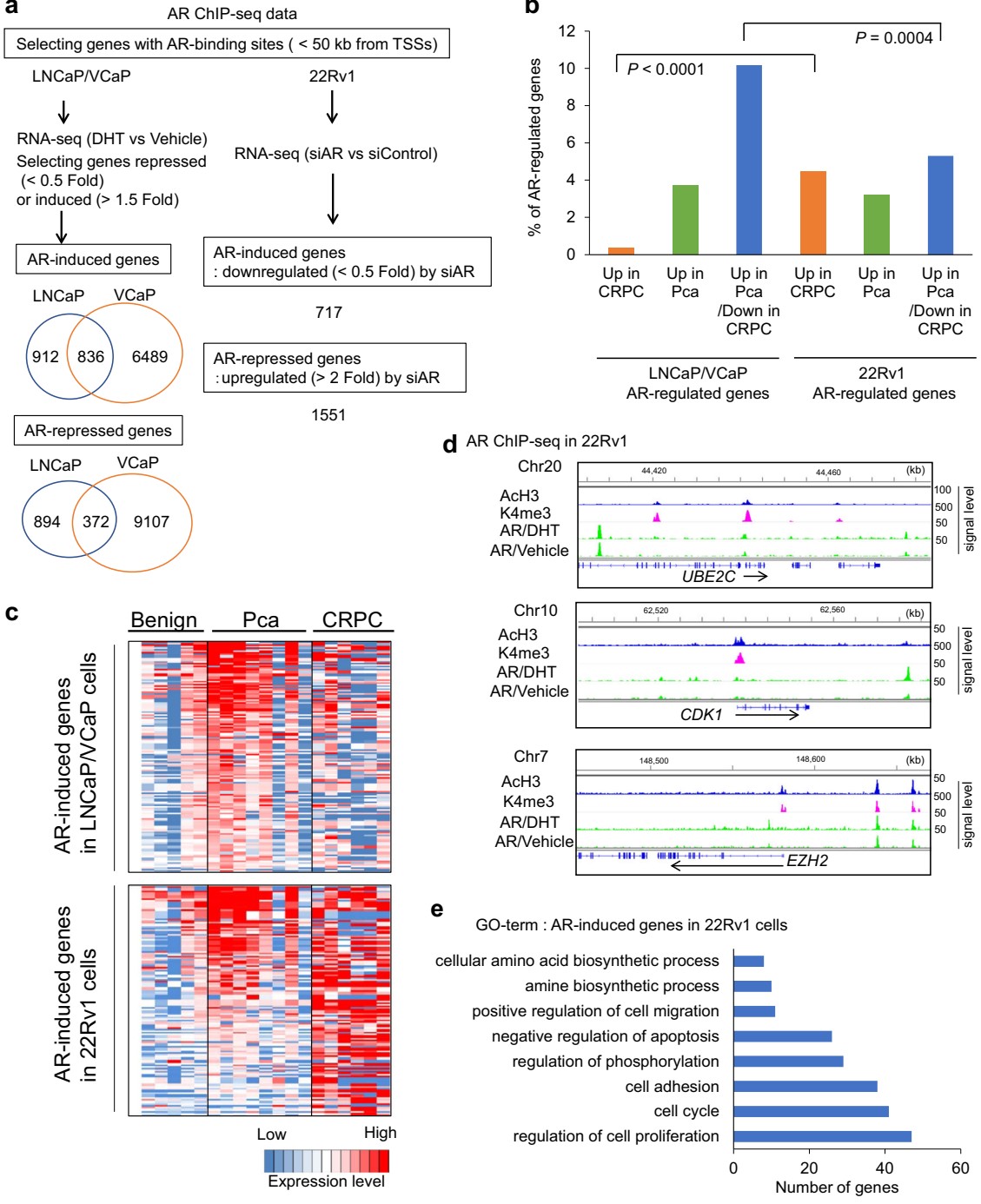

**Fig. 2 AR-regulated gene expression signature in CRPC tissues. a** Workflow for identifying AR-regulated genes in three prostate cancer cell lines. We used AR ChIP-seq and RNA-seq data in three prostate cancer cell lines to determine AR-binding genes induced or repressed by androgen or AR. We selected RefSeq genes with AR-binding sites within 50 kb from transcription start sites (TSSs) as AR-binding genes. For RNA-seq, LNCaP and VCaP cells were treated with DHT 10 nM or vehicle to analyze the regulation by androgen. 22Rv1 cells were treated with siRNA targeting AR (siAR) or control siRNA (siControl) to evaluate the effect of AR on gene expression levels. Thus we selected genes with AR bindings as well as regulated by androgen or AR as AR-target genes. **b** Summary of the expression changes of AR-induced genes. Rate of AR-induced genes in LNCaP/VCaP and 22Rv1 cells overlapped with genes upregulated in CRPC compared with Pca tissues significantly (Up in CRPC), upregulated in Pca compared with benign and downregulated in CRPC compared with Pca tissues significantly (Up in Pca/Down in CRPC), or other genes upregulated in Pca significantly (Up in Pca) are shown. Chi-square test was performed to analyze whether the difference is significant. **c** Expression profile of AR-induced genes upregulated in Pca compared with benign prostate tissues are shown as heatmaps. Top 200 highly expressed AR-induced genes (LNCaP/VCaP and 22Rv1 cells) in Pca tissues are shown. **d** AR regulation of representative CRPC marker genes. AR-binding signals and active histone modification signals (AcH3, K4me3) in 22Rv1 cells obtained by ChIP-seq were summarized. *UBE2C*, *EZH2*, and *CDK1* are shown as representative genes. ChIP-seq signals relative to input control are shown. **e** Gene Ontology (GO) term analysis of AR-regulated genes in 22Rv1 cells.

(Fig. 2b, c) compared to LNCaP/VCaP (0.3%), and they were specifically regulated by AR in 22Rv1 cells (Supplementary Fig. 2d), suggesting the remarkable heterogeneity in the expression of AR-target genes during prostate cancer progression to CRPC. Thus we assumed that AR-regulated signals identified in 22Rv1 cells have similarity in the development of CRPC. Interestingly, among these AR-regulated genes, we found CRPC-specific markers such as *UBE2C*[25]. We also noted that other CRPC marker genes such as *CDK1*[29,30] and *EZH2* were regulated by AR, based on the RNA-seq and ChIP-seq results (Fig. 2d). Within these genes, we observed a significant enrichment of active promoter and enhancer markers (H3K4me3, AcH3) together with AR bindings (Fig. 2d). We found an enrichment of genes associated with "regulation of cell proliferation," "cell cycle," and "cell adhesion" among AR-regulated genes in 22Rv1 cells, which would be important in AR signaling specifically in CRPC (Fig. 2e).

**Identification of AR-regulated lncRNAs induced in CRPC.** Next, to identify the lncRNAs involved in CRPC, we focused on lncRNAs from the transcripts upregulated in CRPC (Type_A or Type_B). Basically, NR_ genes in RefSeq(reads per kilobase of exon per million mapped reads (RPKM) > 5), genes in GENCODE (RPKM > 1), and NONCODE (RPKM > 1) are supposed to be candidates for lncRNAs. However, we noticed that most of the lncRNAs registered in these databases are composed of exons of protein-coding genes and could not be determined to be protein-coding genes or lncRNAs. Therefore, we checked carefully whether the identified transcripts were bona fide lncRNAs or only reflected the detection of exons included in protein-coding genes. Totally, we obtained a total of 91 in Type_A and 72 in Type_B transcripts that were considered as "CRPC-related lncRNAs" (Fig. 3a). We found that some of the previously identified lncRNAs, such as *ARLNC1*[12], *PCAT-1*[21], and *CTBP1-AS*[10], were also upregulated in CRPC compared to benign prostate or localized prostate cancer tissues (Fig. 3b, c).

Recent reports have shown the importance of AR-repressed or AR-induced lncRNAs in CRPC[10–12]. Therefore, we hypothesized that lncRNAs would be useful as novel biomarkers for CRPC diagnosis. Here we compared the sets of AR downstream signals in prostate cancer cells identified by RNA-seq with those obtained as CRPC-related lncRNAs (Fig. 3d, Supplementary Fig. 3). Thus we identified candidates of AR-regulated lncRNAs in CRPC. We then focused on 12 *CRPC-Lnc* RNAs (named as *CRPC-Lncs*) because they were highly expressed in CRPC tissues and their roles in prostate cancer had not been reported previously (Table 1). Interestingly, among these lncRNAs, antisense RNA *HOXC-AS1* (*CRPC-Lnc #9*) located in the HOXC cluster in the vicinity of *HOTAIR* was included. We also observed uncategorized lncRNA expression in the intron of *HOXC4-6* (*HOXC-intron*) to be also overexpressed in CRPC tissues (Fig. 3e). Using ChIP-seq data in 22Rv1 cells, we observed that AR binding and active histone markers were enriched in the promoter or enhancer regions of these lncRNAs (Fig. 3e).

Next, we validated the regulation of 21 lncRNAs by androgen and AR in prostate cancer cells by quantitative reverse transcriptase polymerase chain reaction (qRT-PCR; Fig. 4a, b). We observed both positive and negative regulation of these lncRNAs by AR in 22Rv1 cells. Moreover, using other clinical sample sets, we confirmed the significant upregulation of several lncRNAs (*HOXC-intron*, *CRPC-Lnc #6*, *CRPC-Lnc #9*, and *CRPC-Lnc #11*) as well as *ARLNC1* (Fig. 4c). To explore their functional relevance in CRPC, we further investigated their roles using siRNAs targeting the *CRPC-Lncs*. By MTS (3-(4,5-dimethylthiazol-2-yl)-5-(3-carboxymethoxyphenyl) -2-(4-sulfophenyl)-2H-tetrazolium, inner salt) and cell growth assays,

we observed inhibition of cell proliferation following the silencing of these *CRPC-Lncs* in 22Rv1 and DU145 cells (Fig. 4d). Taken together, these results support the notion that *CRPC-Lncs* are involved in the development and progression of CRPC at a molecular and cellular level.

**Regulation of AR activity by lncRNA-mediated RNA processing.** Recently, several lncRNAs have been reported to be involved in AR activity by modulating AR-epigenetic mechanisms[9,10] and regulating AR expression at both transcriptional and posttranscriptional levels[11,12]. These findings suggest a positive feedback loop between lncRNAs and AR. We first examined whether *CRPC-Lncs* are involved in AR activity by analyzing AR protein expression (Fig. 5a, b). Notably, we found that knockdown of several lncRNAs decreased AR expression, particularly that of AR-V7 at the protein level. Furthermore, qRT-PCR analysis showed that the pre-mRNA of AR was not significantly affected (Fig. 5c), although mature mRNAs were repressed by silencing *CRPC-Lncs* (Fig. 5a). These results indicate that this positive feedback may mainly occur during the RNA processing events such as splicing, exports to the cytoplasm, or maintaining the mRNA stability after splicing. Particularly, we found that knockdown of *CRPC-Lnc #4, #6, #9*, and *#11* repressed AR and AR-V7 expression (Fig. 5a–c) more evidently than the others. Therefore, we analyzed the effects of these lncRNAs by using two siRNAs (Supplementary Fig. 4a–c). In qRT-PCR analysis of AR-target genes and AR luciferase assay, we observed repression of AR-regulated genes and AR activity after silencing of *CRPC-Lncs* (Fig. 5d, e and Supplementary Fig. 4c).

We further examined the role of *CRPC-Lncs* in castration-resistant tumor growth by reducing the expression levels of *CRPC-Lncs* in vivo (Fig. 5f). 22Rv1-derived xenografts were used as a CRPC model implanted into nude mice subcutaneously[16]. Notably, si*CRPC-Lncs* (*Lnc #6-1, Lnc #6-2*, and *Lnc #9*-1) treatment reduced 22Rv1 tumor growth after castration, suggesting that these lncRNAs have a role in CRPC tumor growth. Using western blot analysis, we showed that the expression levels of AR-V7 and full-length AR protein were drastically reduced following the treatments of these si*CRPC-Lncs* (Fig. 5g).

Furthermore, we asked whether AR-mediated regulation of *CRPC-Lncs* is associated with the overexpression in CRPC tissues. Our qRT-PCR analysis showed that *CRPC-Lnc #6* is markedly upregulated following androgen deprivation and highly expressed in the long-term androgen deprivation (LTAD) cells[10,15] derived from LNCaP (Supplementary Fig. 4d). These results suggest that ADT may enhance the expression of *CRPC-Lnc #6* due to its androgen-repressed nature. In contrast, *CRPC-Lnc #9* was not induced by depletion of androgen; however, it was upregulated in 22Rv1 cells compared with LNCaP/VCaP cells. Therefore, we can speculate that *CRPC-Lncs* are upregulated in CRPC tumors by distinct regulatory mechanisms. Consistently, according to The Cancer Genome Atlas datasets, high expression levels of both CRPC-Lnc #6 and #9 are correlated with poor prognosis of prostate cancer patients, suggesting the involvement of their gene expression levels in the disease progression (Supplementary Fig. 5).

**CRPC-Lncs interact with splicing factors for AR splicing.** Finally, we investigated the molecular and cellular mechanisms of *CRPC-Lncs* in the regulation of AR RNA processing. We investigated whether *CRPC-Lncs* target AR through RNA–RNA interaction as observed in the functions of *ARLNC1*[12]. However, we failed to find any sequence similarity between the lncRNAs and mature *AR* mRNA. Therefore, we analyzed the role of lncRNAs in the modulation of splicing factor action, since we have previously shown the importance of overexpression of splicing factors in AR splicing dysregulation[16,31]. First, we

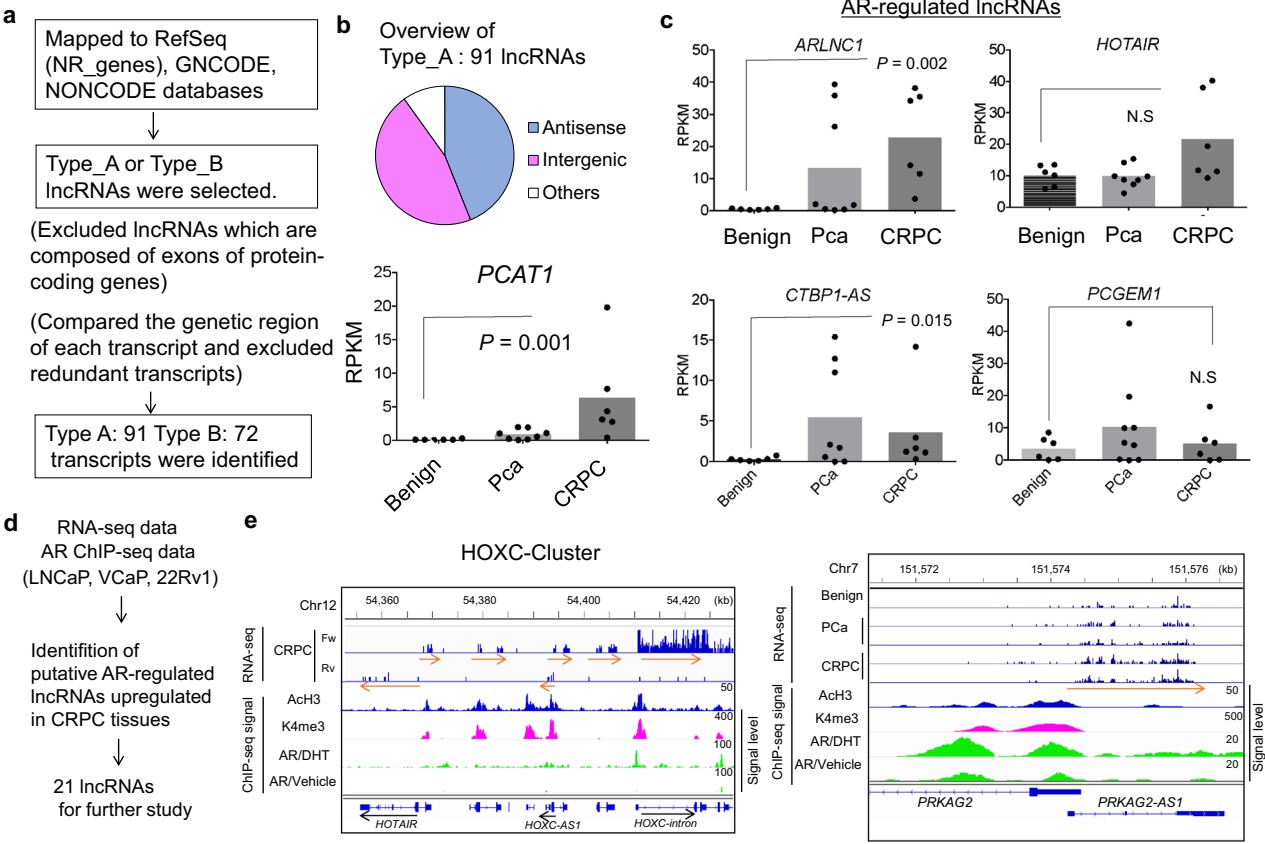

**Fig. 3 Identification of AR-regulated lncRNAs. a** Schematic depicting the analysis to identify lncRNAs upregulated in CRPC tissues. We checked whether obtained lncRNAs in Fig. 1 (Type_A oe Type_B) are truly lncRNA or not by viewing genome browser. Of note, we excluded lncRNAs that are composed of exons of protein-coding genes because they could not be distinguished from protein-coding genes by RNA-seq analysis. A total of 91 Type_A and 72 Type_B candidate lncRNAs were identified. **b** Classification of lncRNAs identified in the present study. LncRNAs were classified to antisense RNA, which is situated at the antisense region of protein-coding genes, intergenic lncRNA, which is not overlapped with protein-coding gene, and others (sense, overlapping, or miRNA precursor). *PCAT1* is shown as a representative lncRNA upregulated in CRPC. *P* value was determined by Mann–Whitney *U* test. **c** Expression levels of representative AR-regulated lncRNAs (*HOTAIR, ARLNC1, CTBP1-AS, PCGEM1*) in RNA-seq analysis. *P* value was determined by Mann–Whitney *U* test. N.S. not significant. **d** Flow chart of the investigation of androgen-regulated lncRNAs upregulated in CRPC tissues. We used RNA-seq in 22Rv1 to obtain AR-repressed (fold>1.5 by siAR) or AR-induced (fold<0.8 by siAR) lncRNAs. We also obtained lncRNAs repressed by androgen (fold<0.8) or induced by androgen (fold>1.5). See also Supplementary Fig. 3. Then we selected 21 transcripts with AR binding in the vicinity (within 10 kb from TSSs) for further studies (Fig. 4a). **e** Representative lncRNAs upregulated in CRPC. RNA-seq results of *CRPC-Lnc #6* (*PRKAG2-AS1*) and *CRPC-Lnc #9* (*HOXC-AS1*) are shown. ChIP-seq signals, AR-binding signals with vehicle or DHT treatment, and active histone modification signals (AcH3, K4me3) in 22Rv1 cells relative to input control are also shown. In *HOXC* locus, other lncRNAs, *HOXC-intron* and *HOTAIR*, were also upregulated in CRPC.

**Table 1 AR-regulated *CRPC-Lnc*s identified in this study.**

| No.: ID | Location | Annotation | Expression level in the RNA-seq study (Average of RPKM) | | |
|---|---|---|---|---|---|
| | | | Benign | Pca | CRPC |
| #1: *NR_039988* | chr22:40431590–40431621 | *n138533* | 13.7 | 12.2 | 31.9 |
| #2: *NONHSAT065996* | chr19:36815838–36822610 | *LINC00665* | 3.0 | 8.4 | 21.8 |
| #3: *ENST00000444958.1* | chr4:53578991–53586518 | *DANCR* | 4.6 | 15.0 | 30.9 |
| #4: *NR_033849* | chr8:144816310–144828507 | *FAM83H-AS1* | 3.1 | 11.0 | 13.2 |
| #5: *NR_110115* | chr7:100951588–100954619 | *LOC101927746* | 0.2 | 0.4 | 7.0 |
| #6: *ENST00000464464.1* | chr7:151574127–151575899 | *PRKAG2-AS1* | 3.3 | 9.6 | 18.0 |
| #7: *ENST00000596366.1* | chr10:104210311–104215640 | *RPARP-AS1* | 9.7 | 18.7 | 27.8 |
| #8: *NR_126416* | chr6:41491633–41515283 | *FOXP4-AS1* | 0.6 | 2.0 | 6.3 |
| #9: *ENST00000512427.1* | chr12:54392806–54393404 | *HOXC-AS1* | 0.1 | 0.0 | 2.2 |
| #10: *NR_132114* | chr16:2204783–2205359 | *SNHG19* | 16.5 | 42.4 | 80.0 |
| #11: *NR_120509* | chr7:1778266–1781946 | *ELFN1-AS1* | 0.0 | 1.1 | 8.0 |
| #12: *NONHSAT068359* | chr19:58823785–58823815 | *ERVK3-1* | 111.5 | 158.9 | 196.75 |

*RNA-seq* RNA-sequencing, *RPKM* reads per kilobase million, *Pca* localized prostate cancer, *CRPC* castration-resistant prostate cancer.

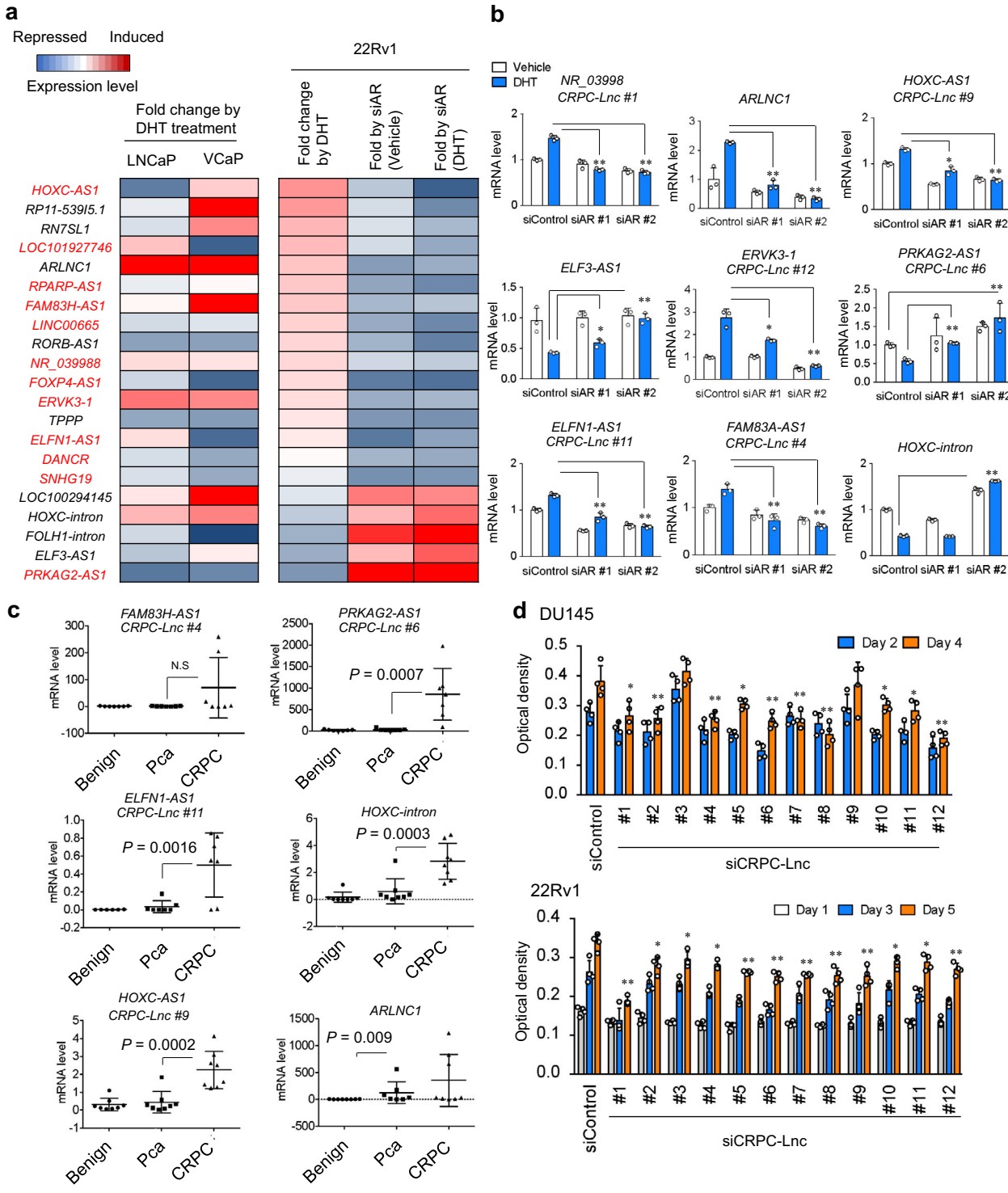

performed RNA immunoprecipitation (RIP) assay to analyze the interaction of *CRPC-Lncs* with the splicing factors. We observed the most significant interaction between *CRPC-Lncs* and U2 small nuclear RNA auxiliary factor 2 (U2AF2)[16,31,32] among the three splicing factors (U2AF2, heterogeneous nuclear ribonucleoprotein A1 (hnRNPA1)[33], and polypyrimidine tract-binding protein-associated splicing factor (PSF))[16] examined (Fig. 6a). RNA pulldown assay showed the interaction of *CRPC-Lnc #6* with U2AF2 but not with PSF, suggesting that this lncRNA-associated splicing factor complex is independent of PSF machinery (Fig. 6b).

We further investigated the role of lncRNAs in splicing machinery to activate AR expression. Surprisingly, we observed that knockdown of *CRPC-Lncs* markedly abrogated U2AF2 association with *AR* pre-mRNA (Fig. 6c). We also showed potential colocalization between *CRPC-Lnc #6* and *AR* pre-mRNA by co-staining in 22Rv1 cells (Fig. 6d). Using a combination of the intron chromosomal expression fluorescence in situ hybridization (FISH) analysis of *AR*, which detect pre-mRNA and *AR* transcriptional loci by staining *AR* intron RNA, and RNA-FISH of *CRPC-Lnc #6*, we observed an overlap of these signals at the *AR* locus (Fig. 6d). Taken together, these findings

**Fig. 4 Regulation of AR-regulated lncRNAs and their roles in CRPC cell growth. a** qRT-PCR validation of AR regulation in AR-positive prostate cancer cells. LNCaP and VCaP cells were treated with 10 nM DHT or vehicle for 24 h. 22Rv1 cells were treated with siControl or siAR (10 nM siAR #2). After 48 h incubation, cells were treated with 10 nM DHT or vehicle for 24 h. Expression levels were determined by qRT-PCR analysis. Average of technical triplicate results is summarized as heatmap. Fold changes by DHT treatment compared with vehicle were shown. Fold change by siAR treatment compared with siControl in the presence or absence of DHT were also shown in the result of 22Rv1 cells. *CRPC-Lnc*s for functional analysis (Table 1) are marked by red letters. **b** qRT-PCR validation of AR regulation in 22Rv1 cells using two types of siAR. Cells were treated with siControl or siAR #1 and #2 (10 nM) for 48 h. siAR#1 targets only AR full length. siAR#2 targets both full length and variants of AR[15]. Then cells were treated with 10 nM or vehicle for 24 h. Expression level of *CRPC-Lnc*s and *HOXC-intron* were determined by qRT-PCR analysis (N = 3). Values represent mean ± S.D. *P < 0.05, **P < 0.01. vs siControl sample treated with DHT. **c** Validation of *CRPC-Lnc*s expression levels in tumor samples by qRT-PCR analysis. Total RNA was extracted from clinically independent samples of benign (N = 8), prostate cancer (N = 8), and CRPC (N = 7 or 8). Expression level of each lncRNA was measured by qRT-PCR. P value was determined by Mann–Whitney U test. Values represent mean ± S.D. **d** Cell proliferation of CRPC model prostate cancer cells was attenuated by knockdown of *CRPC-Lnc*s. DU145 and 22Rv1 cells were treated with siControl or si*CRPC-Lnc*s (10 nM). Cell proliferation was determined by MTS assay (N = 5). Values represent mean ± S.D. *P < 0.05, **P < 0.01 vs siControl.

suggest that *CRPC-Lnc #6* is involved in the splicing events of AR by associating with U2AF2. We revealed that nuclear enrichment of U2AF2 was markedly reduced following knockdown of *CRPC-Lnc #6* by performing immunofluorescence analysis after RNA-FISH (Fig. 6e), whereas the expression level of U2AF2 was not significantly altered by knockdown of *CRPC-Lnc #6* (Fig. 6f). We also showed inhibition of nuclear enrichment of U2AF2 by *CRPC-Lnc #6* knockdown using western blot analysis (Fig. 6g). Moreover, to demonstrate the role of *CRPC-Lnc*s, we established LNCaP cells overexpressing *CRPC-Lnc #6* (Fig. 7a). Consistent with the results of the knockdown, *CRPC-Lnc #6* overexpression increased AR expression and cell growth (Fig. 7b–d). Then we performed RNA-FISH and immunofluorescence analysis in the *CRPC-Lnc #6*-overexpressing cells and showed that nuclear enrichment of U2AF2 was promoted by *CRPC-Lnc #6* over-expression (Fig. 7e, f). Besides, high expression of *CRPC-Lnc #6* in metastatic CRPC tissues and the correlation with *AR* mRNA expression level were shown in public microarray (Grasso et al.[27]) and RNA-seq data (Kumar et al.[34]) (Supplementary Fig. 6). Thus these results suggest that *CRPC-Lnc*s have roles in regulating the splicing factor activity to enhance AR expression by affecting the cellular localization of splicing factors thereby causing dysregulation of splicing (Fig. 7g). We further investigated the effects induced by high expression of *CRPC-Lnc*s on a global gene expression profile using a public gene expression profile (Taylor et al.[28]). We found that high expression levels of *CRPC-Lnc*s are significantly more correlated with gene induction than repression, suggesting that *CRPC-Lnc*s facilitate gene induction of other downstream targets as well as AR in prostate cancer tissues (Supplementary Fig. 7).

## Discussion
Recent studies have shown the usefulness of comprehensive methods to identify new molecular signatures involved in cancer progression. In the present study, we identified important biological signals such as spliceosome and OXPHOS. As we mentioned in the previous reports[16,31], the RNA-binding protein PSF over-expression targeted variety of splicing factors to enhance their expression in prostate cancer cells leading to the dysregulation of the spliceosome in CRPC. A recent work has showed that acti-vated OXPHOS function is associated with docetaxel resistance, suggesting the role of this biological activity in treatment resis-tance[35]. We hypothesize that mitochondrial functions may be enhanced by overexpressing factors of the OXPHOS system.

AR overexpression or amplification is often observed in advanced prostate cancer[3–6]. It is postulated that elucidation of global AR-target genes would facilitate our understanding of the mechanisms underlying progression of CRPC as well as devel-opment of the primary disease. Therefore, we investigated the

genome-wide AR-binding sites in several prostate cancer cells including CRPC models and androgen-dependent prostate cancer cells. A number of reports including ours have used LNCaP cells as a representative model cells for investigating AR-target genes[14,19]. However, this cell line is a model of hormone-sensitive prostate cancer, which is androgen dependent and treatable by AR inhibitors. In this study, we showed that AR-target genes identified in the LNCaP cells are largely upregulated in localized prostate cancer, while they are downregulated in the transition to CRPC. Interestingly, some AR signals identified in the LNCaP cell model such as *COBLL1*, *AMACR*, *ARLNC1*, and *CTBP1-AS* retained high levels of expression in the CRPC tissues. We also compared the expression level of AR-target genes in 22Rv1 cells with genes highly expressed in CRPC specifically. Notably, we found more AR-target genes in 22Rv1 cells among genes upregulated in CRPC tissues compared to LNCaP cells, suggesting that 22Rv1-specific AR signaling is similar to that in clinical CRPC tissues. We then observed that *EZH2* and *CDK1* have multiple AR-binding sites around the promoter and enhancer region in 22Rv1 cells and are regulated by AR. The regulation of *UBE2C*, which is a representative AR-V7 target gene in CRPC[25], was similar. Therefore, it is possible that these important signals in CRPC are widely regulated by AR or AR-V7 directly. Furthermore, our findings indicate comprehensive changes and heterogeneity of AR downstream signals in CRPC tissues in line with past reports[25,36]. Several mechanisms have been suggested, such as the effect of growth factors to facilitate the changes in AR collaborative factors[36].

We then focused on the molecular functions of AR-regulated lncRNAs since the importance of lncRNAs was proposed in prostate cancer progression[9–12]. Among the lncRNAs upregu-lated in CRPC tissues, we found that more than half of them were regulated by AR positively or negatively. Notably, AR-repressed *CRPC-Lnc #6* is markedly upregulated by androgen depletion in AR-positive prostate cancer cells. In contrast, *CRPC-Lnc #9*, which is positively regulated by AR, is not increased by androgen deprivation, whereas upregulated in 22Rv1 cells possibly due to AR overexpression. These observations indicate that several reg-ulatory pathways by AR overexpression in CRPC and LTAD will have an impact on the expression level of the lncRNAs during the clinical course of CRPC. In addition, we observed decreased CRPC tumor growth following silencing of *CRPC-Lnc*s along with reduced expression of AR and AR-V7. Meanwhile, we used two siRNA for silencing *CRPC-Lnc #6* but only one siRNA for *CRPC-Lnc #9*. Further analysis by rescue experiments or overexpression would be required to validate the efficacy of targeting these lncRNAs for the treatment of CRPC.

U2AF2 is a component of the U2 complex in spliceosome to have an important role in *AR* splicing[31,32]. We and other groups revealed that overexpression of various splicing factors including

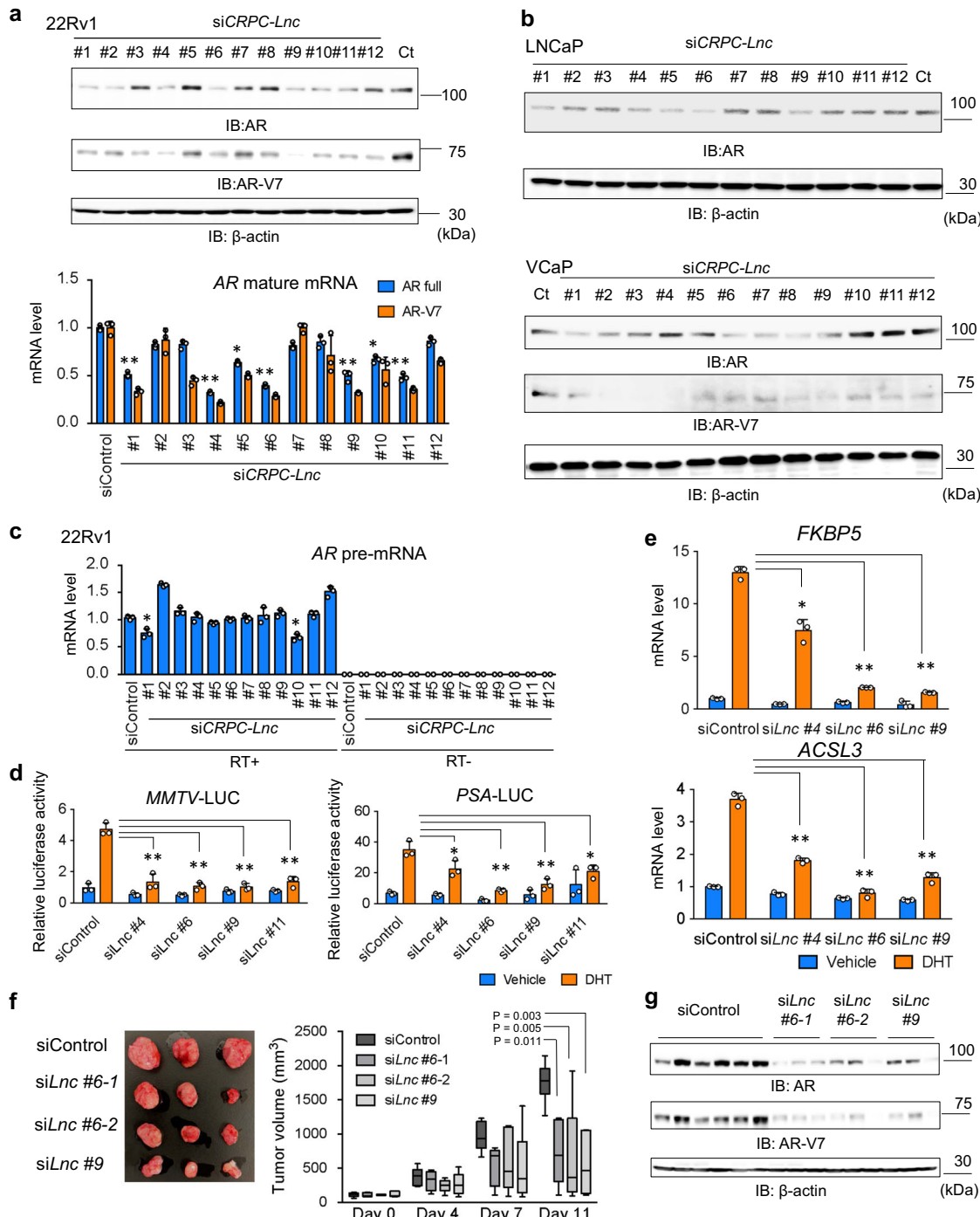

U2AF2 in advanced prostate cancer previously[16,37,38]. Here we demonstrated that the subcellular distribution of U2AF2 was affected in the presence or absence of *CRPC-Lnc #6* using combinational analysis of RNA-FISH and immunofluorescence in prostate cancer cells. These findings in the present study suggested that *CRPC-Lncs* are involved in the dysregulation of splicing efficiency by tuning the activity of splicing machinery in CRPC cells and that such dysregulation of spliceosome complex by lncRNAs would induce other splicing abnormality for other target gene expression levels in advanced prostate cancer. However, other alternative mechanisms regulating the posttranscriptional regulation of *AR* mRNA expression could not be excluded in this study. Further investigation of the underlying mechanisms

of *CRPC-Lncs* associated with gene expression change should be required. Moreover, our analysis of the effects induced by high expression of *CRPC-Lncs* on a global gene expression profile indicates that *CRPC-Lncs* would be involved in gene induction through several regulatory mechanisms, including RNA splicing. We assume that such other downstream targets would have a role in the aggressiveness of prostate cancer tumors. Overall, the present study suggests that targeting the action of these lncRNAs could be a potential strategy to inactivate AR function in CRPC resistant to previous ADT.

In summary, we identified molecular signatures including both protein-coding genes and lncRNAs that are highly expressed in CRPC tissues. Notably, our combinatorial analysis of the global

**Fig. 5 AR-regulated *CRPC-Lnc*s have positive activity for promoting AR splicing and expression. a** Analysis of AR expression by western blot and qRT-PCR analyses in CRPC model cells. 22Rv1 cells were treated with siControl or si*CRPC-Lnc*s for 72 h. Western blot analysis to detect AR protein was performed. β-Actin was used as a loading control. IB immunoblot. Expression level of mature *AR* mRNA (AR full length and AR-V7) was determined by qRT-PCR analysis (*N* = 3). Ct: siControl. **b** Western blot analysis to detect AR protein was performed in other prostate cancer cells. LNCaP and VCaP cells were treated with siControl or si*CRPC-Lnc*s for 72 h. β-Actin was used as a loading control. IB immunoblot, Ct siControl. **c** Pre-mRNA expression of *AR* was not significantly affected by reducing the expression of *CRPC-Lnc*s. Expression level of *AR* pre-mRNA in 22Rv1 cells was analyzed by qRT-PCR using primer spanning *AR* intron 1. RT reverse transcriptase. **d** Luciferase analysis to measure AR activity. 22Rv1 cells were treated with siControl or si*CRPC-Lnc*s for 48 h. Luciferase vectors including AR-binding sites (*PSA*-LUC and *MMTV*-LUC) were transfected. After 24-h incubation, cells were treated with 10 nM DHT or vehicle for 24 h. Cells were lysed, and luciferase activity was measured (*N* = 3). si*Lnc* si*CRPC-Lnc*. **e** AR-target gene induction by androgen is attenuated by *CRPC-Lnc*s silencing. 22Rv1 cells were treated with siControl or si*CRPC-Lnc*s for 48 h. Then cells were treated with 10 nM DHT or vehicle for 24 h. Expression levels of *FKBP5* and *ACSL3* were analyzed by qRT-PCR analysis (*N* = 3). si*Lnc* si*CRPC-Lnc*. Values represent mean ± S.D. **P* < 0.05, ***P* < 0.01. **f** CRPC xenograft tumor growth was inhibited by repressing *CRPC-Lnc*s. Tumor growth of xenografted 22Rv1 cells in castrated nude mice treated with siControl (*N* = 6) or si*CRPC-Lnc*s (si*Lnc*) (*N* = 5) is shown. *P* value was determined by one-way ANOVA, followed by post hoc Dennett's tests. Representative views of tumors in nude mice are shown. **g** Western blot analysis was performed to evaluate AR and AR-V7 expression in biologically independent tumor samples of siControl (*N* = 6), si*CRPC-Lnc* #6 or #9 (*N* = 3). IB immunoblot. *Lnc CRPC-Lnc.*

AR-binding sites with expression profiling indicated that there exist specific molecular networks that facilitate the transition of the AR program. We demonstrated a common underlying mechanism involved in the activation of AR signals by critical CRPC-related lncRNAs that are regulated by androgen. The present study revealed functional modulation of splicing machinery by *CRPC-Lnc*s. It may give a potential opportunity to treat CRPC with activated AR by targeting these *CRPC-Lnc*s.

## Methods

**Subjects and samples**. We obtained prostate cancer samples from surgeries, biopsies, and pathological anatomies conducted at the Jichi Medical University (Tochigi, Japan) and the University of Tokyo (Tokyo, Japan). The study was approved by the Human Genome, Gene Analysis Research Ethics Committee of the Jichi Medical University (No. 18-15), Tokyo Metropolitan Institute of Gerontology (No. 1183), and University of Tokyo (G10044). Written informed consent was obtained from each patient before treatment. This study comprised of six benign prostate and eight localized prostate cancer tissues from eight Japanese patients with prostate cancer. We collected both prostate cancer and benign prostate tissues using laser capture microdissection from prostate tissue sections obtained by radical prostatectomy as described before[13,19]. The age of these patients ranged from 57 to 75 years and the pretreatment serum PSA levels were 4.1–32 ng/mL. A total of six CRPC fresh-frozen tumors were obtained from five patients, who underwent hormone therapy, chemotherapy, and radiotherapy. The clinical characteristics of the CRPC tissues are shown in Supplementary Table 1.

**RNA-sequencing (RNA-seq)**. 22Rv1 cells were transfected with control siRNA (siControl) or siRNA (10 nM) targeting AR (siAR)[15] for 48 h. Following 24 h of treatment of cells with 10 nM DHT or vehicle treatment, cells were collected and total RNA was extracted for RNA-seq analysis. Total RNA was extracted from the benign prostate and prostate cancer tissues as well as CRPC tumor samples and assessed for integrity and concentration using an Agilent 2100 Bioanalyzer system. We confirmed that the RNA integrity numbers of all samples were >9.0. RNA-seq libraries were constructed using the TruSeq RNA Library Preparation Kit v2 (Illumina, San Diego, CA). Sequencing was performed by Illumina HiSeq 2500[16]. For removing rRNA sequences, we used Bowtie 2v. 2.2.6. Mapping to human genome (hg19) was performed using Tophat 2.1.0. Alignments were generated in the SAM format from given single-end reads. For expression analysis, the reads were mapped to the human RefSeq mRNA database or GENCODE/NONCODE database and then we subsequently calculated counts for annotated features, exons, transcripts, or genes. The expression levels of the mapped transcripts were normalized to RPKM to facilitate comparison among different samples. To detect genes highly expressed in CRPC samples, Mann–Whitney *U* test was conducted to identify significantly differentially expressed genes among the benign prostate, prostate cancer, and CRPC samples. Genes with *P* ≤ 0.05 were considered to be significantly upregulated in CRPC. Pathway enrichment in hierarchically clustered gene groups was evaluated using The Database for Annotation, Visualization, and Integrated Discovery (DAVID).

**Cell culture and treatment**. The cell lines used in the present study were obtained from ATCC. Identities of the cells were confirmed by short tandem repeat analyses in 2019 and 2015 (BEX co. Ltd., Tokyo, Japan). All cell lines were grown at 37 °C in a 5% CO$_2$ atmosphere. Also, we routinely checked for mycoplasma contamination using a PCR-based kit, Mycoplasma Detection Kit (Jena Bioscience, Jena, Germany). Additional viral infection was checked using PCR method (ICR monitoring center, Kanagawa, Japan). We maintained stocks of low-passage cells and restarted

our cell culture with a fresh vial at least once a month. VCaP and 293T cells were cultured in Dulbecco's modified Eagle's medium (DMEM) supplemented with 10% fetal bovine serum (FBS), 50 U/mL penicillin, and 50 μg/mL streptomycin. 22Rv1 and LNCaP cells were cultured in RPMI medium supplemented with 10% FBS, 50 U/mL penicillin, and 50 μg/mL streptomycin. LTAD cells were cultured in phenol-red free RPMI medium supplemented with 10% charcoal–dextran-stripped FBS, 50 U/mL penicillin, and 50 μg/mL streptomycin[10]. For androgen treatment, cells were incubated in phenol-red-free RPMI/DMEM medium supplemented with 2.5% charcoal–dextran-stripped FBS for 3 days before androgen stimulation.

**RNA-FISH with immunofluorescence analysis**. Cells were plated on coverslips (Matsunami Glass, Osaka, Japan) and grown for 24–72 h. Cells were fixed using fixation buffer (3.7% formaldehyde) for 10 min. After washing the cells twice with phosphate-buffered saline (PBS), cells were permeabilized by incubating them in 70% ethanol for at least 1 h at 4 °C. After washing twice with PBS, Stellaris RNA-FISH Wash buffer A (SMF-WA1-60, Biosearch Technologies, Hoddesdon, UK) including 10% deionized formamide (Fuji film Wako, Tokyo, Japan) was added to cells and incubated for 5 min. In all, 12.5 μM AR intron and/or *CRPC-Lnc* #6 RNA probe (Supplementary Table 2) labeled with CAL Fluor Red 590 or Quasar 670 in hybridization buffer (90% Stellaris RNA-FISH hybridization buffer (Biosearch Technologies, SMF-HB1-10) and 10% deionized formamide) was added to cells and incubated for 15 h at 37 °C according to the manufacturer's protocol. After washing with Wash buffer A for 30 min at 37 °C, the cells were washed with Wash buffer B (Biosearch Technologies, SMF-WB1-20) for 5 min. To subsequently conduct immunofluorescence, the cells were fixed in 4% paraformaldehyde in PBS (Nacalai) for 10 min at room temperature. After washing the cells thrice in PBS, permeabilization of cells was performed using 0.1% Triton X in PBS for 10 min. Cells were blocked with Blocking one (Nacalai, Tokyo, Japan) for 30 min, and U2AF2 primary antibody was added at a dilution of 1:100 in PBS for 6 h. After washing twice with PBS, the cells were incubated with the secondary antibody at a dilution of 1:1000 in PBS for 1 h in the dark. The cells were then extensively washed thrice with PBS and mounted with mounting medium containing 4′,6-diamidino-2-phenylindole (Nacalai). The images were obtained using confocal microscopy (FV10i, Olympus, Tokyo, Japan). Images were analyzed using the FV10i review system (FV10-ASW ver 2.0, Olympus).

**Reagents and antibodies**. DHT were purchased from Wako (Tokyo, Japan). Rabbit polyclonal anti-AR (ab108341) and anti-AR-V7 (ab198394) antibodies were purchased from Abcam (Cambridge, UK). The rabbit polyclonal anti-H3 (4499) antibody was purchased from Cell Signaling Technology (Danvers, MA). The mouse monoclonal anti-β-actin (A2228) antibody was obtained from Sigma. The mouse monoclonal anti-U2AF2 (MC3), anti-hnRNAPA1 (4B10), and a goat polyclonal anti-GAPDH (V18) antibodies were purchased from Santa Cruz Biotechnology (Dallas, TX).

**Transfection experiments**. Cells were transfected with control siRNA or siRNAs (10 nM) targeting *CRPC-Lnc*s. The siRNAs transfected were as follows: #1: n312690, #2: n507220, #3: n272703, #4-1: n503418, #4-2: n503419, #5: n507156, #6-1: n509295, #6-2: 10620318, #7: n515266, #8: n507972, #9-1: n489630, #9-2: n489629, #10: n513639, #11-1: n508406, #11-2: n50847, and #12: n502900 (Thermo Fisher, Waltham, MA). Two siARs were described before[15]. siAR#1 targets only AR full length. siAR#2 targets both full length and variants of AR[15]. The transfection reagent Lipofectamine RNAiMAX (Thermo Fisher) was used according to the manufacturer's instructions. Cells were incubated for 48–72 h posttransfection to confirm the knockdown. To establish stable cells overexpressing *CRPC-Lnc* #6, we transfected *CRPC-Lnc* #6 pcDNA3.0 or empty vector into LNCaP cells using X-tremeGene HP DNA Transfection Reagent (Roche Applied Science, Penzberg,

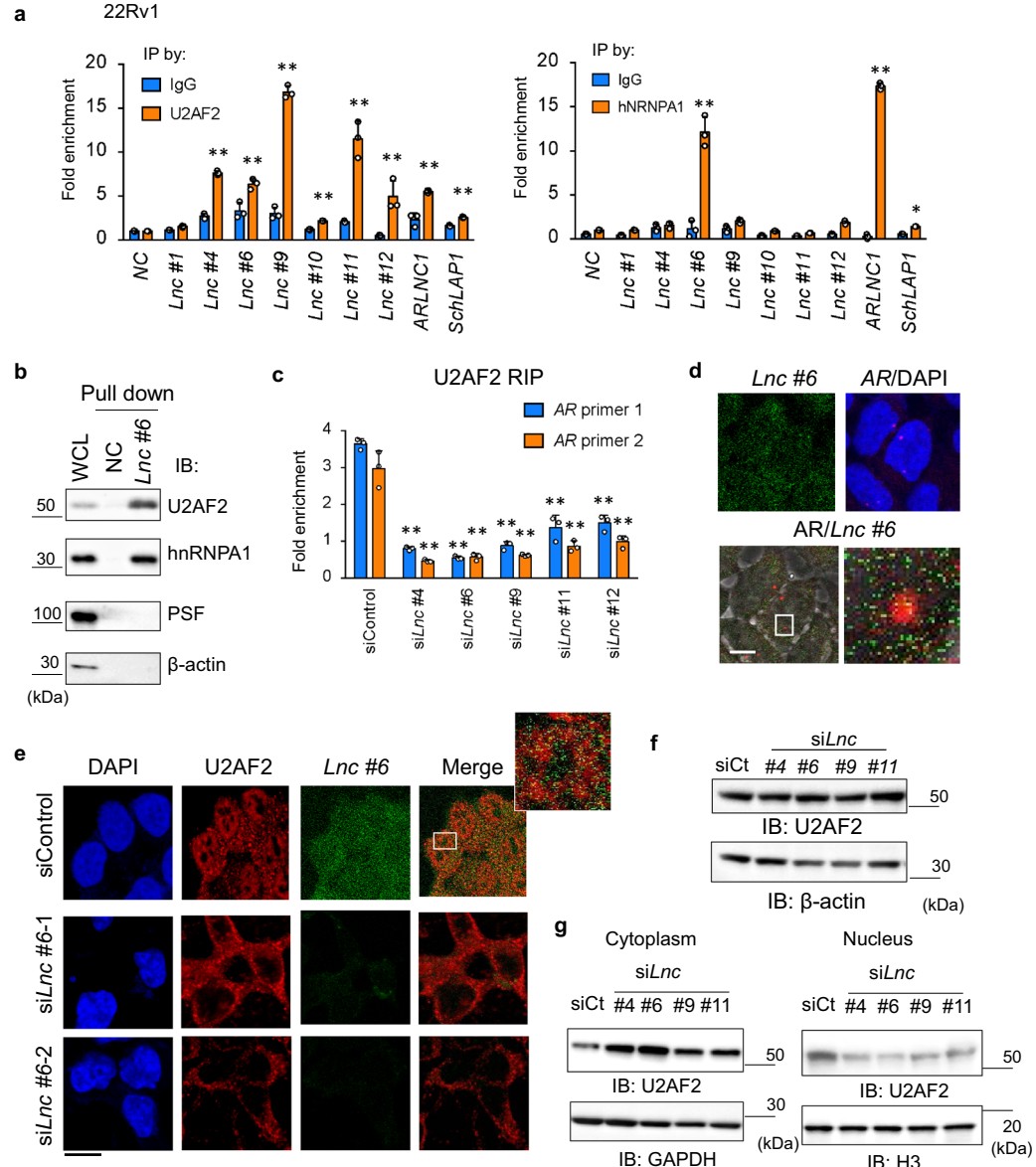

**Fig. 6 Association of *CRPC-Lnc*s with splicing factors responsible for AR splicing machinery. a** RNA immunoprecipitation (RIP) assay showed the interaction of two splicing factors with *CRPC-Lnc*s. 22Rv1 cells were treated with 10 nM DHT or vehicle for 24 h. Cell lysates were immunoprecipitated with normal IgG or specific antibodies targeting splicing factors (U2AF2 and hnRNPA1). Enrichment of each lncRNA was measured by qRT-PCR. *Myoglobin* (*MB*) is used as a negative control (NC). *SchLAP1* and *ARLNC1* were also analyzed (*N* = 3). Values represent mean ± S.D. *P < 0.05, **P < 0.01. *Lnc CRPC-Lnc*. **b** RNA pulldown analysis to analyze the interaction of *CRPC-Lnc #6* with splicing factors. Probes for *CRPC-Lnc #6* fragment (*Lnc #6*) and antisense fragment to *CRPC-Lnc #6* (NC) were prepared. **c** RIP assay to analyze the effects of *CRPC-Lnc*s on the interaction of U2AF2 with AR pre-mRNA (*N* = 3). Cells were treated with siControl or si*CRPC-Lnc*s (#4, #6, #9, #11, #12) for 48 h. Cell lysates were immunoprecipitated with anti-U2AF2 antibody. Enrichment of *AR* pre-mRNA was measured by qRT-PCR using two primers. Values represent mean ± S.D. **P < 0.01. *Lnc CRPC-Lnc*. **d** RNA-FISH analysis to detect *AR* splicing loci and *CRPC-Lnc #6* expression in CRPC. 22Rv1 cells were incubated with RNA probes to detect *AR* intron 3 (*AR*) or *CRPC-Lnc #6* (*Lnc #6*). Bar = 10 µm. **e** Combinational RNA-FISH and immunofluorescence analysis to detect U2AF2 distribution and *CRPC-Lnc #6* (*Lnc #6*) expression. Cells were treated with siControl or si*CRPC-Lnc* (si*Lnc*) #6 for 48 h. Then cells were fixed for RNA-FISH and immunofluorescence analysis. Bar = 10 µm. **f** Western blot analysis to show the expression level of U2AF2 in 22Rv1 cells. Cells were treated with siControl or si*CRPC-Lnc*s (*#4, #6, #9*, and *#11*) for 72 h. **g** Western blot analysis to analyze the distribution of U2AF2 expression in the nucleus and cytoplasm in 22Rv1 cells. Cells were treated with siControl or si*CRPC-Lnc*s (*#4, #6, #9*, and *#11*) for 72 h. H3 was used as a loading control of nuclear fraction of proteins. GAPDH was used as a loading control of cytoplasmic fraction of proteins. IB immunoblot, siCt siControl, *Lnc CRPC-Lnc*.

Germany). Clones expressing *CRPC-Lnc #6* were selected by incubating cells in the presence of G418 (Sigma, St. Louis, MO) at the concentration of 500 µg/mL.

**Western blot analysis**. Nuclear and cytoplasmic proteins were obtained by using hypotonic buffer (20 mM HEPES [pH 7.9], 10 mM KCl, 1 mM EDTA, 1 mM EGTA, 0.65% NP-40, 1 mM dithiothreitol) and RIPA buffer. Protein concentration was calculated by using the BCA Assay Kit (Pierce, Tokyo, Japan). Western blot analysis was performed as described before[16]. Whole-cell extracts were prepared in lysis buffer (50 mM Tris-HCl [pH 8.0], 150 mM NaCl, 1% NP-40, protease inhibitor cocktail (Nacalai). Obtained lysates were loaded on sodium dodecyl sulfate (SDS)–polyacrylamide gels for electrophoresis and subsequently transferred onto

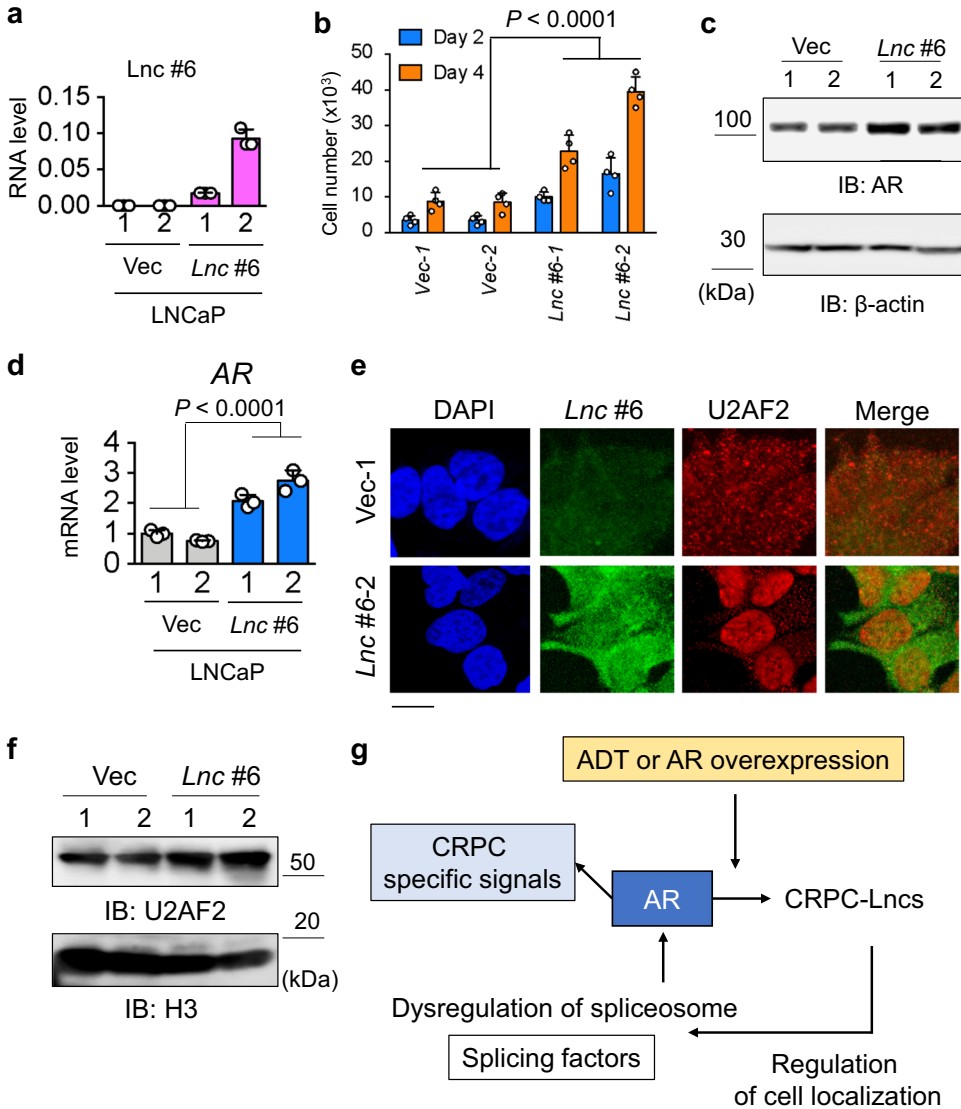

**Fig. 7 Overexpression of *CRPC-Lnc #6* promotes cell growth and AR expression by locating splicing factor U2AF2 in the nucleus. a** Overexpression of *CRPC-Lnc #6* (*Lnc #6*) in LNCaP cells. LNCaP cells stably expressing *CRPC-Lnc #6* were established. Expression level of *CRPC-Lnc #6* was determined by qRT-PCR analysis ($N = 3$). Values represent mean ± S.D. **b** Overexpression of *CRPC-Lnc #6* (*Lnc #6*) promotes cell growth. Cell growth assay by counting viable cells were performed ($N = 4$). Values represent mean ± S.D. Two-way ANOVA was performed to determine the $P$ value. **c** Western blot analysis of AR expression in LNCaP cells overexpressing *CRPC-Lnc #6* (*Lnc #6*). **d** Expression level of AR mRNA was determined by qRT-PCR analysis ($N = 3$). IB immunoblot. Values represent mean ± S.D. Two-way ANOVA was performed to determine the $P$ value. **e** Combinational RNA-FISH and immunofluorescence analysis to detect U2AF2 distribution and *CRPC-Lnc #6* expression. Results of LNCaP cells overexpressing *CRPC-Lnc #6* (*Lnc #6-2*) or control cells (Vec-1) are shown. Bar = 10 μm. **f** Western blot analysis of nuclear U2AF2 expression in LNCaP cells overexpressing *CRPC-Lnc #6*. H3 was used as a loading control of nuclear fraction of proteins. IB immunoblot. **g** Working model of AR and AR-regulated *CRPC-Lnc*s in prostate cancer progression.

Immobilon-P Membranes (Millipore, Billerica, MA, USA). Membranes were incubated with the specific primary antibodies at 4 °C overnight. After incubation with secondary antibodies for 1 h, antibody–antigen complexes were visualized using Western Blotting Detection Reagents (Pierce, Tokyo, Japan).

**RNA isolation and qPCR.** Total RNA was isolated from benign prostate and tumor tissues, as well as from prostate cancer cells, using the Qiagen RNeasy Mini Kit (Qiagen, Venlo, Netherlands). Complementary DNA was produced by PrimeScript (Takara, Kyoto, Japan). qPCR was performed in a StepOne PCR System (Thermo Fisher) using KAPA SYBR Green (Sigma). Gene expression was calculated by ΔΔCt method and normalized to GAPDH at the same condition[16]. Details of the PCR primers are listed in Supplementary Table 3.

**Radioimmunoprecipitation (RIP) assay.** For RIP assay, we used the EZ-magna RIP RNA-binding Protein Immunoprecipitation Kit (Millipore, Burlington, MA) according to the manufacturer's protocol[16]. Briefly, the cell lysates were incubated

with splicing factor antibody coupled IgG magnetic beads at 4 °C overnight. The RNA/antibody complex was washed six times. The RNA was then extracted using ISOGEN and subjected to qRT-PCR. Primer sequences to detect AR intron have been described previously[16].

**RNA pulldown assay.** RNA pulldown was performed as described previously[10] with some modifications. Biotin-labeled *CRPC-Lnc #6* and antisense fragments were prepared using Biotin RNA Labeling Mix (Roche) and T7 RNA polymerase. Biotinylated RNAs were treated with RNase-free DNase (Qiagen), and 10 pmol biotinylated RNA was heated to 60 °C for 10 min and slowly cooled to 4 °C. The RNA was mixed with 100 μg of pre-cleared nuclear extract in RIP buffer supplemented with tRNA (0.1 μg/μL) and incubated at 4 °C for 8 h. A total of 60 μL of washed Streptavidin Agarose beads (Thermo Fisher) was added and incubated for an additional 1 h at 4 °C. The beads were washed five times with RIP buffer and boiled in SDS buffer, and the retrieved proteins were analyzed by western blot analysis.

**Luciferase reporter assay**. LNCaP and 22Rv1 cells were plated in 24-well dishes at a density of $3 \times 10^4$ cells/well in phenol-red-free RPMI medium with 5% charcoal-stripped FBS. After 2 days incubation, the cells were co-transfected with luciferase reporter plasmid (*MMTV* and *PSA*-Luc), Renilla luciferase control, and Tk-pRL vector (Promega, Madison, WI) as a reference using X-tremeGene HP DNA Transfection Reagent (Sigma). After 24 h of incubation, the cells were treated with 10 nM DHT or vehicle (0.1% ethanol) for further 24 h. Luciferase activity was measured using the Dual Luciferase Assay Kit (Promega). The firefly luciferase activity was normalized to the Renilla luciferase activity.

**MTS assay and cell growth assay**. CellTiter 96 Aqueous Kit (Promega), an MTS-based assay, was used to quantify cell vitality and cell growth rate. Cells ($3 \times 10^3$) were plated and cultured in 96-well dishes for MTS assay. For cell growth assay, cells were cultured at $5 \times 10^3$ cells in 24-well plates. Cells were trypsinized and counted using the trypan blue exclusion method. The MTS and cell growth assay were performed in four biological replicates.

**CRPC xenograft model**. Prostate cancer cells, 22Rv1, suspended in 100 μL medium were mixed with 100 μL of Matrigel (BD Biosciences, San Jose, CA) and subcutaneously injected into one side of 20 5-week-old male BALB/c nude mice (CLEA Japan) using a syringe fitted with a 26-G needle. After 7–10 days post-injection, the primary tumor measured approximately 100 mm³. At this point, we performed castration to reduce the androgen concentration in the mice. The mice were randomly divided into four groups. Then 5 μg si*CRPC-Lnc #6-1*, *#6-2*, *#9*, or siControl (BannoNegaCon, Sigma) complexed with Opti-MEM and 15 μL Lipofectamine RNAiMAX (Thermo Fisher) was injected into the tumors 3 times/week for 2 weeks. Tumor volume was calculated using the formula $1/2 \times r1 \times r2^2$ (r1 and r2 (r1 > r2) represent the length and width of the tumor, respectively). Tumors were measured with calipers twice a week. After monitoring the tumor size for 2 weeks, the mice were sacrificed, and the tumor samples were analyzed. Animal care is in accordance with the Tokyo Metropolitan Institute of Gerontology animal experiment guidelines. The ethics committee of animal experiments at the Tokyo Metropolitan Institute of Gerontology approved our study protocol. Tumors were homogenized in ISOGEN (Nippon Gene, Tokyo, Japan) and RIPA buffer with protease inhibitor cocktail (Nacalai) for qRT-PCR and western blot analysis[10,16,17].

**Statistics and reproducibility**. We performed all experiments at least twice and confirmed similar results. *N* values represent technical replicates for qPCR analyses of cancer cell lines but biological and clinical independent samples for other experiments. The data are presented as mean ± SD. In most experiments using cell lines, we used the two-sided Student's *t* test to determine the statistical significance between groups. Significance was defined as $P < 0.05$. qPCR analyses in cell lines were performed in technical replicates. Cell growth assay and luciferase assay were performed in biological replicates. For the assays of stable cell lines, we performed two-way analysis of variance to analyze the significance between the two groups. To determine correlations, we used Spearman's correlation test. Other statistical tests are described in the figure legends. Excel (Microsoft, Redmond, WA) or GraphPad Prism software ver. 6.0 (La Jolla, CA) was used for the statistical analysis.

**Reporting summary**. Further information on research design is available in the Nature Research Reporting Summary linked to this article.

## Data availability

All RNA-seq data have been deposited in the Japanese Genotype-phenotype Archive (JGA)[39] under accession code JGAS00000000198. The Gene Expression Omnibus (GEO) accession numbers for sequence data (RNA-seq and ChIP-seq) used in this study are GSE141806, GSE123565, GSE94577, and GSE82225. The raw data of western blot images are provided in Supplementary Fig. 8; list of all lncRNAs obtained by the current study are summarized in Supplementary Data 1; the source data for the graphs in the main figures and supplementary figures are included in Supplementary Data 2. Other experimental data can be requested from the corresponding author.

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

## Acknowledgements

The authors thank Ms. Noriko Sasaki for her technical assistance, Dr. Kazuhiro Ikeda (Saitama Medical University) for his critical reading of the manuscript, and Former Professor Yukio Homma (University of Tokyo/President Japanese Red Cross Medical Center Tokyo, Japan) for his support in the preparation of clinical samples. This research was supported by grants of P-CREATE (number JP18ck0106194 from AMED, Japan (to S.I.); by grants (to K.T., T.F.) from the JSPS (numbers 17H04334, 18K09128), Japan; and by grants from Takeda Science Foundation, Japan (to S.I., K.T.).

## Author contributions

K.T. designed the study, performed experiments, and analyzed the data. T.F. helped in design the study and provided human resources. Y.S. conducted sequencing. S.I. supervised the study.

## Competing interests

The authors declare no competing interests.
