## [Peer Review File · Communications Biology]

Reviewers' comments:

Reviewer #1 (Remarks to the Author):

Takayama et. al. analyzed their own published RNA-seq and CHIP-seq data to identify lncRNA which may function in CRPC. Once they found candidate lncRNA, they tested their functionality using cell culture and xenograft models and found a possible mechanism explaining their function – regulation of splicing factors.

The manuscript is well written, the data clean and convincing and the claims are well supported by the data. Finding new lncRNA functioning in cancer and is important and can shed a light on the functions of this, relatively little explored, important type of RNA. This story is of interest to readers interested in prostate cancer, lncRNA, splicing regulation and the androgen pathway.

I only have minor comments:

1. Figure 1d – a Venn diagram would be more informative
2. The clinical relevance of the identified lncRNA should be reported using TCGA data and include patient survival vs. lncRNA expression.
3. Do the expression levels of the identified lncRNA, in particular #6 and #9, have a general effect on mRNA splicing? This can be tested, for example, using published RNA-seq data obtained from tissue/cell lines where the lncRNA levels are high vs. low.

Reviewer #2 (Remarks to the Author):

This is a large manuscript, with an extensive amount of work and validation using orthogonal approaches. In it, the authors examine gene and non-coding long RNA (lncRNA) expression between benign, primary, and castration resistant prostate cancer samples and identified AR-regulated lncRNAs. Several of these inhibited CRPC cell growth and inhibited AR transcriptional activity by decreasing AR and AR splice variant levels, which was mediated by the lncRNA interaction with U2AF2, a splicing factor. This work is convincing and would be of interest to the community.

Overall, this is a really nice manuscript. There are two points that should be addressed to improve it. For one, the discussion is a little bit long and portions on oxidative phosphorylation could be easily removed to tighten up the manuscript, as the authors do little functional analysis of OXPHOS in their studies (this manuscript is large enough; these studies aren't required and would in fact be better in a separate manuscript). Also, the title wording could be improved. Perhaps "Long non-coding RNAs dysregulate (promote/support/alter etc) AR splicing in advanced prostate cancer?"

Reviewer #3 (Remarks to the Author):

In this manuscript, Takayama and colleagues study the role of lncRNAs in regulating splicing of AR in castration resistant prostate cancer cell line and patient tissues. Results from their RNA-seq and ChIP-seq analysis informed further analysis of few lncRNAs like lncRNA6 in regulating splicing events associated with splicing factor U2AF2 specifically. The authors hypothesize a role of lncRNAs in directly regulating splicing machinery to affect expression of AR and AR variants to further impact tumor growth in castration resistant models. The data presented in the manuscript support some of the conclusions (such as the shift in localization of U2AF2 in presence or absence of lncRNAs). However, as outlined below, there are other major conclusions that are not supported by the data. This work will likely be of interest to researchers investigating the role of lncRNAs in cancer especially advanced prostate cancer and also to researchers studying various splicing mechanisms that regulate gene

expression in disease models.

Major Comments:

1. In Figure 1D, the comparison of the small RNA-seq cohort in this study with prior datasets is not statistically valid or robust. The comparison with Varambally dataset is not appropriate as Varambally is an even smaller cohort of 4 CRPC vs. 15 primary. The authors should test against the Taylor MSKCC dataset of 19 CRPC vs ~130 primaries). The study makes interesting observations in patient tissue samples, but the results would be more convincing if the study had more power through more specimens.
2. The approach used to integrate RNA-seq and ChIP-seq data in 2a is unclear. Differentially expressed genes in the "vicinity" of an AR binding site was used as the threshold for identifying "AR regulated genes". What is meant by "vicinity"? It is impossible to conclude that these genes are AR regulated just because they are located near an AR binding site.
3. The manuscript overall suffers from providing insufficient details in figure legends (especially for figures 1 and 2) and inconsistent or incomplete labeling and typographical errors. Several times, information regarding biological or technical replicates or statistical tests and criteria is inadequate. For example, missing lane information on the western blot in figure 5g and different scales in figure 6e.
4. Data shown in Figure 2b do not truly represent its explanation in corresponding lines 147-151 on page 10. This is not a statistically valid or robust way to compare gene expression datasets from different cell line models.
5. Figure 2d should illustrate ChIP-seq peaks by visualizing BAM or BIGWIG files, not by illustrating BED file intervals. Same for Figure 3d.
6. The lncRNA compendium used for analysis in Figure 3a is inadequate because it eliminates lncRNAs that overlap with annotated genes. Many important lncRNAs reside within annotated genes.
7. Figure 3c is not a statistically valid or robust way to compare gene expression datasets.
8. RT-PCR data in Figure 5c do not robustly support splicing regulation as an underlying mechanism. There are many alternative interpretations of the data presented. The study generally assumes that AR variants is the only mechanism of regulating expression of AR and its enhanced downstream signaling in CRPC.
9. The rationale for selection of lncRNAs 4, 6, and 9 for follow-up is unclear
10. The study convincingly shows that loss of lncRNA6 leads to reduced tumor growth in vivo by using 2 separate siRNAs. However the data for lncRNA9 is only derived from a single siRNA. This raises concern about cherry-picking of positive data.
11. In Figure 6b, the significance and specificity of lncRNA interactions with U2AF2 are unclear. Since the lncRNAs appear to have biological effects in AR-null DU145 cells (Figure 4D), it is likely that biological effects of the putative lncRNA/U2AF2 interactions are completely independent of AR. The data illustrated do not rigorously establish lncRNA 6 is an important AR splicing regulator.

Minor Comments:

1. Lines 41-42, need to specify that this is deaths from cancer.
2. The anatomic sites of CRPC metastases obtained from biopsy/autopsy are not provided
3. In figure 3c, the VCaP+LNCaP together in response to DHT (grey) is missing.
4. Use of 'oxidation' instead of 'oxidative' phosphorylation on at least 2 separate occasions in the text and similar instances of typographical errors in the manuscript.
5. Figures like fig. 4a need to be labeled appropriately for the reader.

Answer to Reviewer #1

Takayama et. al. analyzed their own published RNA-seq and CHIP-seq data to identify lncRNA which may function in CRPC. Once they found candidate lncRNA, they tested their functionality using cell culture and xenograft models and found a possible mechanism explaining their function – regulation of splicing factors. The manuscript is well written, the data clean and convincing and the claims are well supported by the data. Finding new lncRNA functioning in cancer and is important and can shed a light on the functions of this, relatively little explored, important type of RNA. This story is of interest to readers interested in prostate cancer, lncRNA, splicing regulation and the androgen pathway.

We thank the reviewer for the insightful and constructive comments. We revised the manuscript according to the comments.

1. Figure 1d – a Venn diagram would be more informative

As suggested by the reviewer, we modified the figure 1d to show the overlap between genes upregulated in our study and those upregulated in other cohorts. In the revised manuscript, we replaced the cohort of Varambally et al. with that of Taylor et al. to respond to the comment of reviewer #3.

2. The clinical relevance of the identified lncRNA should be reported using TCGA data and include patient survival vs. lncRNA expression.

We analyzed the expression of CRPC-Lnc #6 and #9 in TCGA prostate cancer cohort using a web tool (Supplementary Fig.5). We added survival curves showing the importance of these lncRNAs to predict the prognosis of prostate cancer patients.

Page 14, Ln.220-223

Consistently, according to the Cancer Genome Atlas (TCGA) data sets, high expressions of both CRPC-Lnc #6 and #9 are correlated with poor prognosis of prostate cancer patients, suggesting the involvement of their gene expressions in the disease progression (Supplementary Figure 5).

3. Do the expression levels of the identified lncRNA, in particular #6 and #9, have a general effect on mRNA splicing? This can be tested, for example, using published RNA-seq data obtained from tissue/cell lines where the lncRNA levels are high vs. low.

It is important to analyze the specific effect of these CRPC-Lncs on global gene expression and splicing as pointed out by the reviewer. To address the concern, we analyzed the difference of gene expression profiles between in samples with high expression of CRPC-Lnc #6 and #9 and low by using public microarray data (Taylor et al.) obtained from prostate cancer tissues, as suggested by the reviewer. We then assume the positive effect of these lncRNA expressions on a subset of mRNA expressions possibly by regulating the splicing (Supplementary Fig.7).

Page 16, Ln.262-265

We found that high expressions of CRPC-Lncs are significantly more correlated with gene induction than repression, suggesting that *CRPC-Lncs* facilitate gene induction of other downstream targets as well as AR in prostate cancer tissues (Supplementary Figure 7).

Page 21, Ln.321-325

Moreover, our analysis of the effects induced by high expression of *CRPC-Lncs* on a global gene expression profile indicates that *CRPC-Lncs* would be involved in gene induction through several regulatory mechanisms including RNA splicing. We assume such other downstream targets would have a role in the aggressiveness of prostate cancer tumors.

Answer to Reviewer #2

This is a large manuscript, with an extensive amount of work and validation using orthogonal approaches. In it, the authors examine gene and non-coding long RNA (lncRNA) expression between benign, primary, and castration resistant prostate cancer samples and identified AR-regulated lncRNAs. Several of these inhibited CRPC cell growth and inhibited AR transcriptional activity by decreasing AR and AR splice variant levels, which was mediated by the lncRNA interaction with U2AF2, a splicing factor. This work is convincing and would be of interest to the community.

We thank the reviewer for the insightful and constructive comments. We modified the Discussion part more concise and changed the title.

Overall, this is a really nice manuscript. There are two points that should be addressed to improve it. For one, the discussion is a little bit long and portions on oxidative phosphorylation could be easily removed to tighten up the manuscript, as the authors do little functional analysis of OXPHOS in their studies (this manuscript is large enough; these studies aren't required and would in fact be better in a separate manuscript).

As suggested by the reviewer, we shortened the paragraph of discussion on OXPHOS and made the discussion part more concise by removing several sentences.

Also, the title wording could be improved. Perhaps “Long non-codings RNAs dysregulate (promote/support/alter etc) AR splicing in advanced prostate cancer?”

As pointed by the reviewer, we changed the title to “Long non-coding RNAs dysregulate splicing activity to promote androgen receptor pathway in advanced prostate cancer”.

Answer to Reviewer #3

In this manuscript, Takayama and colleagues study the role of lncRNAs in regulating splicing

of AR in castration resistant prostate cancer cell line and patient tissues. Results from their RNA-seq and ChIP-seq analysis informed further analysis of few lncRNAs like lncRNA6 in regulating splicing events associated with splicing factor U2AF2 specifically. The authors hypothesize a role of lncRNAs in directly regulating splicing machinery to affect expression of AR and AR variants to further impact tumor growth in castration resistant models. The data presented in the manuscript support some of the conclusions (such as the shift in localization of U2AF2 in presence or absence of lncRNAs). However, as outlined below, there are other major conclusions that are not supported by the data. This work will likely be of interest to researchers investigating the role of lncRNAs in cancer especially advanced prostate cancer and also to researchers studying various splicing mechanisms that regulate gene expression in disease models.

We thank the reviewer for the critical, insightful and inspiring comments. We revised the manuscript as suggested by the reviewer by adding more description in figure legends and discussion. We also modified some figures to make figures more valid and support the conclusion.

1. In Figure 1D, the comparison of the small RNA-seq cohort in this study with prior datasets is not statistically valid or robust. The comparison with Varambally dataset is not appropriate as Varambally is an even smaller cohort of 4 CRPC vs. 15 primary. The authors should test against the Taylor MSKCC dataset of 19 CRPC vs ~130 primaries). The study makes interesting observations in patient tissue samples, but the results would be more convincing if the study had more power through more specimens.

To address the concern raised by the reviewer, we downloaded the microarray data of Taylor et al. and replaced the result of Varambally dataset with it. In addition, we selected genes upregulated or downregulated in metastatic CRPC tissues in two cohorts (Grasso et al. and Taylor et al.). and showed the comparison by Venn diagrams as instructed by another reviewer (reviewer #1). The overlap between genes identified in our results with these cohorts was statistically significant by chi-square test. We added these results in revised Figure 1d.

2. The approach used to integrate RNA-seq and ChIP-seq data in 2a is unclear. Differentially expressed genes in the “vicinity” of an AR binding site was used as the threshold for identifying “AR regulated genes”. What is meant by “vicinity”? It is impossible to conclude that these genes are AR regulated just because they are located near an AR binding site.

“Genes in the vicinity of ARBSs” means genes closest to ARBSs. We deleted the expression of “genes in the vicinity” from the figure. We identified AR-target genes by using both ChIP-seq and RNA-seq data. These AR-targets are not only closest to ARBSs but also regulated by androgen treatment or AR knockdown at mRNA level, suggesting the transcriptional regulation of these genes

by AR.

Page 9, Ln.126-129

We used three AR ChIP-seq data to select AR-binding genes, which were closest genes to AR-binding sites in androgen-dependent type prostate cancer cells (LNCaP and VCaP) and CRPC model cells (22Rv1) (Fig. 2a). In addition, to determine the regulation of gene expression by androgen, RNA-seq studies for these cells were performed.

3. The manuscript overall suffers from providing insufficient details in figure legends (especially for figures 1 and 2) and inconsistent or incomplete labeling and typographical errors. Several times, information regarding biological or technical replicates or statistical tests and criteria is inadequate. For example, missing lane information on the western blot in figure 5g and different scales in figure 6e.

To respond to the reviewer's comment, we added more details in figure legends including information about statistical tests and criteria of significance. We added more details such as the information about statistical tests, replicates of experiments in "statistical analysis" in the Methods part. We corrected some labels and typographical errors of figures. We showed lane information in Figure 5g, and uniformed scales in Figure 6e.

Page 31, Ln.503-512

Statistical analysis

We performed all experiments at least twice and confirmed similar results. The data are presented as the mean \pm SD. In most experiments using cell lines, we used the two-sided Student's t-test to determine the statistical significance and calculate P values between groups. qPCR analysis was performed in technical replicates. Cell growth assay and luciferase assay were performed in biological replicates. Significance was defined as $P < 0.05$. For the *in vitro* proliferation assay of the stable cell lines, we performed two-way analysis of variance (ANOVA) to determine the significance between two groups. To analyze correlations, we used Spearman's correlation test. Excel (Microsoft, Redmond, WA) or GraphPad Prism software ver. 6.0 (La Jolla, CA) was used for the statistical analysis.

4. Data shown in Figure 2b do not truly represent its explanation in corresponding lines 147-151 on page 10. This is not a statistically valid or robust way to compare gene expression datasets from different cell line models.

To address the concern, we performed chi-square test to analyze whether the difference is significant

or not statistically. We added the P-value to the figure and showed the validity.

5. Figure 2d should illustrate ChIP-seq peaks by visualizing BAM or BIGWIG files, not by illustrating BED file intervals. Same for Figure 3d.

As suggested by the reviewer, we showed ChIP-seq signals by using WIG files.

6. The lncRNA compendium used for analysis in Figure 3a is inadequate because it eliminates lncRNAs that overlap with annotated genes. Many important lncRNAs reside within annotated genes.

As pointed out by the reviewer, important lncRNAs are distributed in the annotated gene loci. We excluded transcripts composed of exons of protein-coding genes because we could not distinguish these lncRNAs from protein-coding genes by the result of RNA-seq analysis. However, we included lncRNAs expressed in intron or antisense regions of annotated genes, which would be important. We modified the description in Figure 3a to show our investigation more clearly.

Page 11, Ln.159-166

Basically, NR_ genes in RefSeq (**RPKM > 5**), genes in GENCODE (**RPKM > 1**) and NONCODE (**RPKM > 1**) are supposed to be candidates for lncRNAs. However, we noticed that most of the lncRNAs registered in these databases **are composed of exons of** protein-coding genes and **could not be determined to** be protein-coding genes or lncRNAs. Therefore, we checked carefully whether the identified transcripts were *bona fide* lncRNAs or only reflected the detection of exons included in protein-coding genes. Totally, we obtained **a total of 91 in Type_A and 72 in Type_B transcripts** that were considered as ‘CRPC-related lncRNAs’ (Fig. 3a).

7. Figure 3c is not a statistically valid or robust way to compare gene expression datasets.

As pointed out by the reviewer, this figure did not show statistical result of comparison. Therefore, we determined to move this figure to supplementary figure 3. In addition, because the content of this figure is overlapped with that of Supplementary Figure 3b and 3d, we modified this figure and summarized the numbers of genes regulated by AR in 22Rv1 (Supplementary Figure 3e) simply.

8. RT-PCR data in Figure 5c do not robustly support splicing regulation as an underlying mechanism. There are many alternative interpretations of the data presented. The study generally assumes that AR variants is the only mechanism of regulating expression of AR and its enhanced downstream signaling in CRPC.

Although most of these CRPC-Lncs were involved in the expression of AR/AR-V7 mRNA levels

(Figure 5a), we observed that pre-mRNA levels are not affected by silencing of CRPC-Lncs (Figure 5c). This finding suggested that AR expression levels are regulated at post-transcriptional level including splicing. As suggested by the reviewer, this result does not support splicing regulation is the only mechanism in regulating AR expression by CRPC-Lncs. Thus, we modified some sentences and modified discussion to describe other mechanisms and downstream signaling induced by lncRNAs.

Page 13, Ln. 198-200

These results indicate that this positive feedback may mainly occur during the RNA processing events such as **splicing, exports to the cytoplasm or maintaining the mRNA stability after splicing.**

Page 20, Ln.318-321

However, other alternative mechanisms regulating the posttranscriptional regulation of AR mRNA expression could not be excluded in this study. Further investigation of the underlying mechanisms of CRPC-Lncs associated with gene expression change should be required.

9. The rationale for selection of lncRNAs 4, 6, and 9 for follow-up is unclear

To respond to the comment, we added description about the reason to focus on these CRPC-Lncs.

Page 13, Ln. 200-203

Particularly, we found that knockdown of CRPC-Lnc #4, 6, 9, 11 repressed AR and AR-V7 expression (Fig. 5a-c) more evidently than the others. Therefore, we analyzed the effects of these lncRNAs by using two siRNAs (Supplementary Fig. 4a-c).

10. The study convincingly shows that loss of lncRNA6 leads to reduced tumor growth in vivo by using 2 separate siRNAs. However, the data for lncRNA9 is only derived from a single siRNA. This raises concern about cherry-picking of positive data.

We performed this xenograft experiment by using three siRNAs (siLnc6-1, siLnc6-2, siLnc9-1) with control siRNA, but did not pick up only positive data. As suggested by the reviewer, it would be ideal to use more siRNAs to validate the effect of silencing these lncRNAs. Unfortunately, we could not use more because of the limited capacity to conduct the experiment. Meanwhile, as demonstrated in the supplementary figure 4, we obtained a similar effect of another siRNA targeting Lnc #9 on cell growth and AR activity, suggesting that knockdown of these lncRNAs by using different sequence of siRNA could have a similar impact.

11. In Figure 6b, the significance and specificity of lncRNA interactions with U2AF2 are unclear. Since the lncRNAs appear to have biological effects in AR-null DU145 cells (Figure 4D), it is likely that biological effects of the putative lncRNA/U2AF2 interactions are completely independent of AR. The data illustrated do not rigorously establish lncRNA 6 is an important AR splicing regulator.

In the previous studies, U2AF2 is shown to be an important splicing regulator for AR splicing process (ref.16, 32). We additionally demonstrated the localization of U2AF2 is modulated by CRPC-Lnc #6 in this study, suggesting that CRPC-Lnc #6 might have a role in tuning the activity of U2AF2. Our experimental studies such as RIP assay, RNA pulldown assay, and RNA FISH demonstrated the interaction of Lnc #6 with U2AF2. As for AR splicing, U2AF2 interaction with pre-mRNA of AR was reduced by silencing of Lnc #6 (Fig. 6c). However, I agree with the reviewer in the point that this interaction have more effects independent of AR because U2AF2 would have more target genes as a splicing regulator. Therefore, we analyzed the effect of CRPC-Lncs on gene expression by using public microarray data as suggested by reviewer #1. We added description about the role of CRPC-Lnc #6 in regulating other target genes in Discussion part.

Page 16, Ln.260-265

We further investigated the effects induced by high expression of CRPC-Lncs on a global gene expression profile using a public gene expression profile (Taylor et al28). We found that high expressions of CRPC-Lncs are significantly more correlated with gene induction than repression, suggesting that CRPC-Lncs facilitate gene induction of other downstream targets as well as AR in prostate cancer tissues (Supplementary Figure 7).

Page 21, Ln. 321-325

our analysis of the effects induced by high expression of *CRPC-Lncs* on a global gene expression profile indicates that *CRPC-Lncs* would be involved in gene induction through several regulatory mechanisms including RNA splicing. We assume such other downstream targets would have a role in the aggressiveness of prostate cancer tumors.

We further analyzed the correlation of CRPC-Lnc #6 with AR expression by using public dataset to support the conclusion. We found this lncRNA expression is significantly associated with AR in CRPC tissues in two cohorts, indicating the importance of CRPC-Lnc #6 as a regulator of AR expression (Supplementary Fig. 6).

Page 16, Ln. 255-258

Besides, high expression of *CRPC-Lnc #6* in metastatic CRPC tissues and the correlation with

AR mRNA expression level were shown in public microarray (Grasso et al.²⁷) and RNA-seq data (Kumar et al.³⁵) (Supplementary Figure 6).

Minor Comments:

1. Lines 41-42, need to specify that this is deaths from cancer.

We corrected as pointed out by the reviewer (Page 4, Ln. 38).

2. The anatomic sites of CRPC metastases obtained from biopsy/autopsy are not provided

We added this information in the figure legend (Ln. 685).

3. In figure 3c, the VCaP+LNCaP together in response to DHT (grey) is missing.

We removed this panel to respond to the major comment (#7) of this reviewer.

4. Use of ‘oxidation’ instead of ‘oxidative’ phosphorylation on at least 2 separate occasions in the text and similar instances of typographical errors in the manuscript.

We corrected “oxidation” to “oxidative”.

5. Figures like fig. 4a need to be labeled appropriately for the reader.

We modified the label more understandable for the reader.

Reviewers' comments:

Reviewer #3 (Remarks to the Author):

The revisions incorporated into this manuscript have improved presentation and accuracy. As outlined below, there are two outstanding points that have not been addressed satisfactorily by the authors.

Major Comments:

1. In Figure 1D, the comparison of the small RNA-seq cohort in this study with prior datasets is not statistically valid or robust. The comparison with Varambally dataset is not appropriate as Varambally is an even smaller cohort of 4 CRPC vs. 15 primary. The authors should test against the Taylor MSKCC dataset of 19 CRPC vs ~130 primaries). The study makes interesting observations in patient tissue samples, but the results would be more convincing if the study had more power through more specimens.

The authors' response to this comment is satisfactory.

2. The approach used to integrate RNA-seq and ChIP-seq data in 2a is unclear. Differentially expressed genes in the "vicinity" of an AR binding site was used as the threshold for identifying "AR regulated genes". What is meant by "vicinity"? It is impossible to conclude that these genes are AR regulated just because they are located near an AR binding site.

The authors' response to this comment raises an additional concern. The authors have defined an "AR-binding gene" as the gene closest to an AR binding site. Because AR can bind to genomic sites that are very distant from any annotated genes, it is standard practice to impose distance cutoffs. The authors do not appear to have imposed such cutoffs in this manuscript. It would be more accurate and consistent with literature on AR chromatin binding studies to define an AR-binding gene as any gene +/- 10 kb or +/- 50 kb from an AR binding site. Any genes further than this distance should not be classified as an AR-binding gene.

3. The manuscript overall suffers from providing insufficient details in figure legends (especially for figures 1 and 2) and inconsistent or incomplete labeling and typographical errors. Several times, information regarding biological or technical replicates or statistical tests and criteria is inadequate. For example, missing lane information on the western blot in figure 5g and different scales in figure 6e.

The authors' response to this comment is satisfactory.

4. Data shown in Figure 2b do not truly represent its explanation in corresponding lines 147-151 on page 10. This is not a statistically valid or robust way to compare gene expression datasets from different cell line models.

The authors' response to this comment is satisfactory.

5. Figure 2d should illustrate ChIP-seq peaks by visualizing BAM or BIGWIG files, not by illustrating BED file intervals. Same for Figure 3d.

The authors' response to this comment is satisfactory.

6. The lncRNA compendium used for analysis in Figure 3a is inadequate because it eliminates lncRNAs

that overlap with annotated genes. Many important lncRNAs reside within annotated genes.

The authors' response to this comment is satisfactory.

7. Figure 3c is not a statistically valid or robust way to compare gene expression datasets.

The authors' response to this comment is satisfactory.

8. RT-PCR data in Figure 5c do not robustly support splicing regulation as an underlying mechanism. There are many alternative interpretations of the data presented. The study generally assumes that AR variants is the only mechanism of regulating expression of AR and its enhanced downstream signaling in CRPC.

The authors' response to this comment is satisfactory.

9. The rationale for selection of lncRNAs 4, 6, and 9 for follow-up is unclear

The authors' response to this comment is satisfactory.

10. The study convincingly shows that loss of lncRNA6 leads to reduced tumor growth in vivo by using 2 separate siRNAs. However the data for lncRNA9 is only derived from a single siRNA. This raises concern about cherry-picking of positive data.

The authors' response to this comment is not satisfactory. These results obtained using a single siRNA reagent are used to support major conclusions in the study. It is well-known that siRNA knock-down can have off-target effects, which is why best practices require the use of two independent siRNAs and/or rescue of any knock-down phenotype by re-expression of the knock-down target. Since neither of these best practices were utilized, off-target effects can't be ruled out. Given that the authors are unable to conduct these experiments, the text of the abstract, results, and discussion should be modified to tone down these major conclusions regarding lncRNAs 6 and 9.

11. In Figure 6b, the significance and specificity of lncRNA interactions with U2AF2 are unclear. Since the lncRNAs appear to have biological effects in AR-null DU145 cells (Figure 4D), it is likely that biological effects of the putative lncRNA/U2AF2 interactions are completely independent of AR. The data illustrated do not rigorously establish lncRNA 6 is an important AR splicing regulator.

The authors' response to this comment is satisfactory.

Minor Comments:

1. Lines 41-42, need to specify that this is deaths from cancer.

The authors' response to this comment is satisfactory.

2. The anatomic sites of CRPC metastases obtained from biopsy/autopsy are not provided

The authors' response to this comment is satisfactory.

3. In figure 3c, the VCaP+LNCaP together in response to DHT (grey) is missing.

The authors' response to this comment is satisfactory.

4. Use of 'oxidation' instead of 'oxidative' phosphorylation on at least 2 separate occasions in the text and similar instances of typographical errors in the manuscript.

The authors' response to this comment is satisfactory.

5. Figures like fig. 4a need to be labeled appropriately for the reader

The authors' response to this comment is satisfactory.

Answers to the Reviewer #3

2. The approach used to integrate RNA-seq and ChIP-seq data in 2a is unclear. Differentially expressed genes in the “vicinity” of an AR binding site was used as the threshold for identifying “AR regulated genes”. What is meant by “vicinity”? It is impossible to conclude that these genes are AR regulated just because they are located near an AR binding site.

The authors’ response to this comment raises an additional concern. The authors have defined an “AR-binding gene” as the gene closest to an AR binding site. Because AR can bind to genomic sites that are very distant from any annotated genes, it is standard practice to impose distance cutoffs. The authors do not appear to have imposed such cutoffs in this manuscript. It would be more accurate and consistent with literature on AR chromatin binding studies to define an AR-binding gene as any gene +/- 10 kb or +/- 50 kb from an AR binding site. Any genes further than this distance should not be classified as an AR-binding gene.

A. To respond to the comment of this reviewer, we selected AR-binding genes, which have AR-binding sites within 50 kb from transcription start sites (Figure 2a). Accordingly, we reanalyzed the expression level of AR-regulated genes in our RNA-seq data and GO-term of AR-regulated genes (Figure 2b, 2c and 2e). Although the numbers of AR-regulated genes were changed, the overall results obtained by this reanalysis were in line with those of previous analysis. Therefore, we revised some description about this analysis in the Results section [Page 10, Ln. 142, 155] and figure legends [Page 38, Ln. 712].

Page 10, Ln. 142

However, most of them (10.2%) were downregulated in the CRPC tissues compared to the prostate cancer tissues (Fig.2b and 2c). In contrast, among AR-regulated genes in 22Rv1 cells, the number of genes upregulated in prostate cancer tissues (8.4%) was lower ($P = 0.001$ by chi-square test) than that in LNCaP/VCaP (13.8%) (Fig. 2b).

Page 10, Ln. 155

We found an enrichment of genes associated with ‘regulation of cell proliferation’, ‘cell cycle’, and ‘cell adhesion’ among AR-regulated genes in 22Rv1 cells, which would be important in AR signaling specifically in CRPC (Fig. 2e).

Page 38, Ln. 712

We selected RefSeq genes **with AR-binding sites within 50 kb from transcription start sites (TSSs)** as AR-binding genes.

10. The study convincingly shows that loss of lncRNA6 leads to reduced tumor growth in vivo by using 2 separate siRNAs. However the data for lncRNA9 is only derived from a single siRNA. This raises concern about cherry-picking of positive data.

The authors' response to this comment is not satisfactory. These results obtained using a single siRNA reagent are used to support major conclusions in the study. It is well-known that siRNA knock-down can have off-target effects, which is why best practices require the use of two independent siRNAs and/or rescue of any knock-down phenotype by re-expression of the knock-down target. Since neither of these best practices were utilized, off-target effects can't be ruled out. Given that the authors are unable to conduct these experiments, the text of the abstract, results, and discussion should be modified to tone down these major conclusions regarding lncRNAs 6 and 9.

A. We appreciated this reviewer for giving us a useful suggestion for future analyses. We agree with the reviewer's comment about the validity of the xenograft experiment. To address the concern raised by the reviewer, we toned down the conclusion by deleting or changing some words associated with the xenograft experiment such as the efficacy of using lncRNAs for the treatment of CRPC in Abstract [Page 3, Ln 29, 32], Results [Page 13, Ln. 209], and Discussion [Page 20, Ln. 308] sections. In addition, we added a description about the limitation of this experiment in the Discussion [Page 20, Ln. 310] section.

Page 3, Ln. 29

Notably, silencing of two lncRNAs (*CRPC-Lnc #6: PRKAG2-AS1* and *#9: HOXC-AS1*) **alleviated** CRPC tumor growth, showing repression of AR and AR variant expression.

Page3, Ln. 32

our investigation highlights a new cluster of lncRNAs which could serve as AR regulators **as well as potential biomarkers in CRPC.**

Page 13, Ln. 209

Notably, si*CRPC-Lncs* (*Lnc #6-1, #6-2* and *Lnc #9-1*) treatment **reduced** 22Rv1 tumor growth

after castration, suggesting that these lncRNAs have a role in CRPC tumor growth.

Page 20, Ln. 308

In addition, we observed decreased CRPC tumor growth following silencing of *CRPC-Lncs* along with reduced expression of AR and AR-V7. Meanwhile, we used two siRNA for silencing CRPC-Lnc #6 but only one siRNA for CRPC-Lnc #9. Further analysis by rescue experiments or overexpression would be required to validate the efficacy of targeting these lncRNAs for the treatment of CRPC.